# Gauging U(1) symmetry in $(2+1)d$ topological phases

**Meng Cheng[1] and Chao-Ming Jian[2]**

**1** Department of Physics, Yale University, New Haven, CT 06511-8499, USA
**2** Department of Physics, Cornell University, Ithaca, NY 14853, USA

## Abstract

We study the gauging of a global U(1) symmetry in a gapped system in (2+1)d. The gauging procedure has been well-understood for a finite global symmetry group, which leads to a new gapped phase with emergent gauge structure and can be described algebraically using the mathematical framework of modular tensor category (MTC). We develop a categorical description of U(1) gauging in a MTC, taking into account the dynamics of U(1) gauge field absent in the finite group case. When the ungauged system has a non-zero Hall conductance, the gauged theory remains gapped and we determine the complete set of anyon data for the gauged theory. On the other hand, when the Hall conductance vanishes, we argue that gauging has the same effect of condensing a special Abelian anyon nucleated by inserting $2\pi$ U(1) flux. We apply our procedure to the SU(2)$_k$ MTCs and derive the full MTC data for the $\mathbb{Z}_k$ parafermion MTCs. We also discuss a dual U(1) symmetry that emerges after the original U(1) symmetry of an MTC is gauged.



## 1  Introduction

For a quantum many-body system with a global symmetry $G$, coupling to a background $G$ gauge field is often a particularly effective way to probe the symmetry property of the ground state. When the gauge field becomes dynamical, a new quantum phase with $G$ gauge structure emerges. This gauging construction/approach has played a key role in the recent advances in the classifications of symmetry-protected topological (SPT) phases and symmetry-enriched topological (SET) phases [1–5], and more generally has illuminated many new relations between seemingly different quantum field theories [6].

Roughly speaking, gauging a $G$ symmetry modifies the theory in two ways: first, new excitations carrying $G$ fluxes are introduced. Second, the Gauss's law is imposed and only $G$-invariant states are kept in the Hilbert space. For a finite $G$, at low energy the $G$ gauge field stays completely flat, and as a result the local dynamics is essentially unaffected by gauging. Equivalently, $G$ fluxes in this case are gapped, localized objects. If the ungauged system is gapped, gauging a finite group symmetry again leads to a gapped phase.

When the system before gauging is gapped and thus a $G$ symmetry-enriched topological phase, the gauging procedure can be systematically described within the mathematical framework of $G$-crossed braided tensor category [1]. Mathematically, a (2+1)d gapped phase in a bosonic system is described by a modular tensor category (MTC) $\mathcal{C}$, also known as the algebraic theory of anyons [7]. All universal aspects of a gapped phase with $G$ symmetry can be captured algebraically by $G$ action on the MTC $\mathcal{C}$ [1]. Then from this information, there is a well-defined procedure to produce a new MTC representing the gauged system, as described in Ref. [1].

Gauging a continuous symmetry is however fundamentally different from the finite group case, due to dynamical considerations. In this note we will be focusing on U(1) symmetry. It is well-known that in general a compact U(1) Maxwell gauge theory is completely confined at low energy due to the proliferation of instantons [8]. Here by instanton we mean an insertion of $2\pi$ magnetic flux, which is a local operator. In other words, the theory describes a completely trivial phase. This is already dramatically different from the finite group case, where the "garden variety" gauge theory is always deconfined, with intrinsic topological order. On the other hand, if the U(1) gauge theory comes with a Chern-Simons (CS) term, then the theory is also gapped but with nontrivial topological order.

---

[1]Modulo some ambiguity corresponding to stacking $G$ SPT phases

The purpose of this note is to provide a purely algebraic formulation for gauging U(1) symmetry in a (2+1)d gapped phase. U(1) symmetry in an MTC $\mathcal{C}$ can be captured by two pieces of data: an Abelian anyon $v$, the vison, encoding fractional charges carried by the anyons, and the Hall conductance $\sigma_H$. With this triplet $\mathcal{C}, v$ and $\sigma_H$, when $\sigma_H \neq 0$ we construct a new MTC $\mathcal{D}$ (i.e. with all $F$- and $R$-symbols explicitly determined) which corresponds to the gauged theory, by formalizing the notion of flux attachment. Mathematically, to obtain the categorical data of the new MTC $\mathcal{D}$, we first introduce an infinite category whose anyons are labeled by all combinations of the anyon labels of the original MTC $\mathcal{C}$ and the compatible U(1) charges. Fusion rules of anyons in the infinite category are derived from those of the original MTC $\mathcal{C}$ and the addition of U(1) charges. Braiding-related data in the infinite category can be obtained from the those of $\mathcal{C}$ plus extra contributions from Aharonov-Bohm phases caused by flux attachment. The MTC $\mathcal{D}$, namely the gauged theory, can be obtained by condensing the a transparent anyon, which is directly associated with the vison $v \in \mathcal{C}$, in this infinite category. We show that the new gauged MTC $\mathcal{D}$ is also mathematically equivalent to $\mathcal{C} \boxtimes U(1)_{-s^2 \sigma_H}\big|_{(v, s\sigma_H)}$ which is derived from an anyon condensation in $\mathcal{C} \boxtimes U(1)_{-s^2 \sigma_H}$. Here, $s$ is the minimal integer such that $s$ copies of the vison $v$ fuse into a trivial anyon. More detailed explanation of this notation will be given later. When $\sigma_H = 0$, we argue that gauging U(1) amounts to condensing the vison $v$, which can be understood as the consequence of Polyakov's mechanism for confinement. We also discuss in this note an emergent $U(1)_{\text{dual}}$ symmetry of the gauged theory $\mathcal{D}$ in both the case with $\sigma_H \neq 0$ and the case with $\sigma_H = 0$.

## 2 Gauging U(1) in an Abelian topological phase

Let us first study gauging of an Abelian topological phase. It is known that any Abelian topological order in (2+1)d can be represented by a $U(1)^N$ CS theory [9,10]. When the (2+1)d Abelian topological order has a global U(1) symmetry, we can couple the $U(1)^N$ CS theory to the background U(1) gauge field associated with the global U(1) symmetry:

$$\mathcal{L} = -\frac{K_{IJ}}{4\pi} a_I d a_J + \frac{t_I}{2\pi} A d a_I, \tag{1}$$

where $a_{I=1,2,\dots,N}$ denote the dynamical U(1) gauge fields in the $U(1)^N$ CS theory, and $A$ is the background U(1) gauge field. Here, $K$ is an $N \times N$ non-degenerate symmetric integer matrix. In this paper, we only focus on topological orders in bosonic systems. Hence, the diagonal entries of $K$ are all required to be even. The coupling between $a_I$ and $A$ is determined by the charge vector $\mathbf{t} = (t_1, t_2, \dots, t_N)^\top$. We will assume that the charge vector is primitive, i.e. $\gcd(t_1, t_2, \dots, t_N) = 1$. Formally integrating out $a_I$'s one readily sees that the system has a Hall conductance $\sigma_H = \mathbf{t}^\top K^{-1} \mathbf{t}$.

A quasiparticle or an anyon in the Abelian topological order described by Eq. (1) can be labeled by a $N$-component integer vector $\mathbf{l} \in \mathbb{Z}^N$ that specifies the gauge charge of this anyon under the $U(1)^N$ gauge group. Two anyons $\mathbf{l}$ and $\mathbf{l}'$ are topologically equivalent if $\mathbf{l} - \mathbf{l}' = K\mathbf{m}$ for some $\mathbf{m} \in \mathbb{Z}^N$. It implies that quasiparticles that are labeled by $K\mathbf{m}$ for any $\mathbf{m} \in \mathbb{Z}^N$ are trivial, namely they are local excitations and are topologically equivalent to the vacuum. The fusion rule of the anyons in theory Eq. (1) is simply given by the addition of the gauge charges under the $U(1)^N$ gauge group. All anyons in this theory are Abelian. The topological twist factor of the anyon $\mathbf{l}$ is given by $\theta_{\mathbf{l}} = e^{i\pi \mathbf{l}^\top K^{-1} \mathbf{l}}$. The braiding statistics between the Abelian anyons $\mathbf{l}$ and $\mathbf{l}'$ is given by $M_{\mathbf{l}\mathbf{l}'} = e^{i2\pi \mathbf{l}^\top K^{-1} \mathbf{l}'}$. In addition, one can show that the chiral central charge $c_-$ of the system is given by the signature of the matrix $K$, i.e. $c_- = \text{sig}\, K = r_+ - r_-$, where $r_\pm$ is the number of positive/negative eigenvalues of $K$.

The coupling to the U(1) background gauge field $A$ in Eq. (1) specifies how the global U(1)

symmetry acts on this Abelian topological order. In particular, it implies, via the equation of motion, that the anyon $\mathbf{l}$ carries charge $Q_{\mathbf{l}} = \mathbf{t}^{\mathsf{T}} K^{-1} \mathbf{l}$ under the global U(1) symmetry. Also, we observe that the insertion of a $2\pi$ flux of $A$ induces a specific anyon, the vison, which is given by $v = \mathbf{t}$. Notice that identity that

$$M_{v\mathbf{l}} = e^{\mathrm{i}2\pi Q_{\mathbf{l}}}, \tag{2}$$

for any anyon $\mathbf{l} \in \mathbb{Z}^N$. Physically, it means that we can reinterpret the braiding statistics between the vison $v$ and the anyon $\mathbf{l}$ as the Aharonov-Bohm phase between the $2\pi$-flux of $A$ (that is the vison) and the U(1) symmetry charge $Q_{\mathbf{l}}$ carried by the anyon $\mathbf{l}$. Notice that the charge $Q_{\mathbf{l}}$ is fractional if and only if the braiding statistics $M_{v\mathbf{l}}$ is non-trivial.

When we gauge the U(1) global symmetry in the theory Eq. (1), we promote $A$ to a dynamical U(1) gauge field. The resulting theory is a U(1)$^{N+1}$ CS theory with a new $K$-matrix:

$$\tilde{K} = \begin{pmatrix} K & -\mathbf{t} \\ -\mathbf{t}^{\mathsf{T}} & 0 \end{pmatrix}, \tag{3}$$

whose determinant is

$$\det \tilde{K} = \mathbf{t}^{\mathsf{T}} K^{-1} \mathbf{t} \det K = \sigma_H \det K. \tag{4}$$

Clearly, the physics of the gauged theory crucially depends on whether $\sigma_H$ vanishes or not. We will discuss the two cases separately.

## 2.1 $\sigma_H \neq 0$

Let us first consider $\sigma_H \neq 0$ which implies that $\det \tilde{K} \neq 0$. Therefore, the gauged system is a gapped system whose Abelian topological order is described by the new $K$-matrix $\tilde{K}$ in Eq. (3).

First, we observe that with $\sigma_H \neq 0$, the signature of $\tilde{K}$ is

$$\mathrm{sig}(\tilde{K}) = \mathrm{sig}(K) - \mathrm{sgn}\,\sigma_H. \tag{5}$$

This follows from the following identity:

$$W^{\mathsf{T}} \tilde{K} W = \begin{pmatrix} K & 0 \\ 0 & -\sigma_H \end{pmatrix}, \quad W = \begin{pmatrix} \mathbf{1} & K^{-1}\mathbf{t} \\ 0 & 1 \end{pmatrix}, \tag{6}$$

and the fact that the signature is invariant under invertible similarity transformation. Since the signature of the $K$-matrix is the chiral central charge of the topological phase, we find that

$$c'_- = c_- - \mathrm{sgn}\,\sigma_H, \tag{7}$$

where $c_- = \mathrm{sig}(K)$ is the chiral central charge before gauging the U(1) symmetry and $c'_- = \mathrm{sig}(\tilde{K})$ is that of the gauged system.

While the gauged theory is basically given by the $\tilde{K}$ matrix in Eq. (3), in the following we analyze the anyon content of theory, in particular how they are related to anyons in the theory before gauging, in a manner that does not rely on the explicit U(1)$^N$ CS field theory construction. As will be shown later, the result can be easily adopted to more general topological phases.

Anyons in the gauged theory can be represented by $(N + 1)$-component integer vectors $\begin{pmatrix} \mathbf{l} \\ q \end{pmatrix}$, where $\mathbf{l} \in \mathbb{Z}^N$ labels the anyons in the original ungauged theory (i.e. gauge charges

under the gauge fields $a_{I=1,2,...,N}$) and $q \in \mathbb{Z}$ can be understood as an additional charge under $A$ attached to the anyon $\mathbf{l}$. We can represent all local excitations in the following form

$$\tilde{K}\begin{pmatrix} \mathbf{m} \\ n \end{pmatrix} = \begin{pmatrix} K\mathbf{m} - n\mathbf{t} \\ -Q_{K\mathbf{m}} \end{pmatrix}, \tag{8}$$

where $\mathbf{m} \in \mathbb{Z}^N$ and $n \in \mathbb{Z}$. After gauging the U(1) symmetry, there are two kinds of local excitations. The first kind is of the form $\begin{pmatrix} K\mathbf{m} \\ -Q_{K\mathbf{m}} \end{pmatrix}$. In particular, we note that the additional charge under $A$ attached is $-Q_{K\mathbf{m}}$, exactly canceling the the original $A$-charge $Q_{K\mathbf{m}}$ carried by $K\mathbf{m}$ to form a charge-neutral object under $A$. The other type of local excitations is generated by $\begin{pmatrix} \mathbf{t} \\ 0 \end{pmatrix}$, which is the vison of the original theory, with no additional $A$-charge attached.

A useful result is the inverse of the new $K$-matrix:

$$\tilde{K}^{-1} = \begin{pmatrix} K^{-1} - \sigma_H^{-1}K^{-1}\mathbf{t}\mathbf{t}^{\mathsf{T}}K^{-1} & -\sigma_H^{-1}K^{-1}\mathbf{t} \\ -\sigma_H^{-1}\mathbf{t}^{\mathsf{T}}K^{-1} & -\sigma_H^{-1} \end{pmatrix}. \tag{9}$$

Therefore, in the theory after we gauged the U(1) symmetry, the topological twist factor for the anyon $\mathbf{p} = \begin{pmatrix} \mathbf{l} \\ q \end{pmatrix}$ is then given by

$$\theta_{\mathbf{p}} = \exp\left(i\pi \mathbf{l}^{\mathsf{T}}K^{-1}\mathbf{l} - i\pi\sigma_H^{-1}(q + Q_{\mathbf{l}})^2\right) = \theta_{\mathbf{l}} e^{-\frac{i\pi}{\sigma_H}(q+Q_{\mathbf{l}})^2}. \tag{10}$$

Here, $\theta_{\mathbf{l}} = e^{i\pi \mathbf{l}^{\mathsf{T}}K^{-1}\mathbf{l}}$ is the topological twist factor of the anyon $\mathbf{l}$ in the ungauged theory. Also, we recognize $q + Q_{\mathbf{l}}$ as the total charge under $A$ carried by the excitation $\mathbf{p} = \begin{pmatrix} \mathbf{l} \\ q \end{pmatrix}$, and the additional phase factor $e^{-\frac{i\pi}{\sigma_H}(q+Q_{\mathbf{l}})^2}$ in can be understood as Aharonov-Bohm-like phase from flux attachment for the gauge field $A$. The braiding statistics $M_{\mathbf{p}\mathbf{p}'}$ of the gauged theory between the anyons $\mathbf{p} = \begin{pmatrix} \mathbf{l} \\ q \end{pmatrix}$ and $\mathbf{p}' = \begin{pmatrix} \mathbf{l}' \\ q' \end{pmatrix}$ has a similar structure:

$$M_{\mathbf{p}\mathbf{p}'} = e^{i2\pi\mathbf{p}^{\mathsf{T}}\tilde{K}^{-1}\mathbf{p}'} = M_{\mathbf{l}\mathbf{l}'} e^{-\frac{i2\pi}{\sigma_H}(q+Q_{\mathbf{l}})(q'+Q_{\mathbf{l}}')}, \tag{11}$$

which is the product of the braiding statistics $M_{\mathbf{l}\mathbf{l}'}$ between anyons $\mathbf{l}$ and $\mathbf{l}'$ in the ungauged theory and an extra phase factor $e^{-\frac{i2\pi}{\sigma_H}(q+Q_{\mathbf{l}})(q'+Q_{\mathbf{l}}')}$ due to flux attachment of $A$.

## 2.2 $\sigma_H = 0$

When the Hall conductance is zero, the new $K$-matrix has a vanishing determinant, namely $\det\tilde{K} = 0$. In this case, the corresponding U(1)$^{N+1}$ CS theory by itself is not a valid description of a topological phase. To see why this is the case, when $\det\tilde{K} = 0$, there exist null vectors $\mathbf{v}$ such that $\tilde{K}\mathbf{v} = \mathbf{0}$. It is easy to show that the null space (over real numbers) is generated by the following vector:

$$\begin{pmatrix} K^{-1}\mathbf{t} \\ 1 \end{pmatrix}. \tag{12}$$

One can multiply a certain integer to make it integral, and we call the minimal such integral vector $\mathbf{v}_0$. It is then always possible to find an invertible transformation $W$ such that by a change of variable $a = Wa'$,

$$W^{\mathsf{T}}\tilde{K}W = \begin{pmatrix} 0 & 0 \\ 0 & \tilde{K}' \end{pmatrix}, \tag{13}$$

where $\tilde{K}'$ is non-degenerate. Denote the gauge field corresponding to the 0 corner by $a_1'$, and the corresponding vector $\mathbf{e}_1' = (1, 0, 0, \dots)$. Because there is no Chern-Simons term for $a_1'$, on general grounds we should include a Maxwell term for $a_1'$. Then due to Polyakov mechanism $a_1'$ becomes confined, and because $a_1'$ has no (topological) coupling to the remaining gauge fields we can safely ignore $a_1'$. The "remaining" non-degenerate $K$-matrix $\tilde{K}'$ should describe the resulting topological order after the gauging of U(1).

While such invertible transformation $W$ always exists, the explicit expression and the resulting $\tilde{K}'$ are often quite cumbersome and not particularly enlightening. In the following we use an alternative formulation of Abelian CS theory to sidestep the need to find $\tilde{K}'$ explicitly. In this formulation, we think of the theory described by a non-degenerate $K$-matrix $K$ as an integral lattice $L^K = \mathbb{Z}^{\dim K}$ where each vector represents an excitation. The lattice is equipped with a symmetric bilinear form given by $K^{-1}$. Then we form the quotient $L^K / L_{\text{loc}}^K$ where $L_{\text{loc}}^K = \{K\mathbf{l} | \mathbf{l} \in L^K\}$, which is a finite Abelian group with a non-degenerate quadratic form. First, we observe that finding the resulting topological order $\tilde{K}'$ is the same as determining the sub-lattice orthogonal to the vector $\mathbf{v}_0$. More explicitly, a vector $\mathbf{l}$ in the original basis becomes $W^{-1}\mathbf{l}$ after basis transformation, which is orthogonal to $\mathbf{e}_1' = W^{-1}\mathbf{v}_0$ if and only if $\mathbf{l}^\mathsf{T}\mathbf{v}_0 = 0$. Therefore, a general vector $\begin{pmatrix} \mathbf{x} \\ n \end{pmatrix}$ is orthogonal to $\mathbf{v}_0$ if and only if $\mathbf{x}^\mathsf{T} K^{-1}\mathbf{t} = -n$. In other words, under the gauge field $A$, the charge of the anyon corresponding to $\mathbf{x}$ in the ungauged theory has to be an integer, which can be made zero by attaching $-n$ local $A$-charges and then survives the confinement of $a_1'$. Physically, this is what one expects from "condensing" the vison (i.e. $2\pi$ flux) in the original ungauged theory, as the condensation should confine all anyons which braid non-trivially with $\mathbf{t}$, i.e. carrying fractional charge under $A$. It is useful to notice that the orthogonality condition uniquely determines the $(N + 1)$-th component $n$ in terms of the first $N$ components $\mathbf{x}$. In other words, the sublattice of $L^{\tilde{K}}$ orthogonal to $\mathbf{v}_0$ is actually isomorphic to a subspace of the original lattice $L^K$ in which all vectors have integer charge: $L_0^K = \{\mathbf{l} | \mathbf{l} \in L^K, Q_\mathbf{l} \in \mathbb{Z}\}$.

Now that we have determined the space orthogonal to $\mathbf{v}_0$, it is still necessary to quotient out the local excitations. It is easy to see that the two kinds of local excitations, one in the form of $\begin{pmatrix} K\mathbf{m} \\ -Q_{K\mathbf{m}} \end{pmatrix}$ and the other generated by $\begin{pmatrix} \mathbf{t} \\ 0 \end{pmatrix}$, are both in the orthogonal space $L_0^K$. In fact, they only differ from $L_{\text{loc}}^K$ by the inclusion of $\mathbf{t}$. So in the end we find that the topological order after the gauge the U(1) symmetry corresponds to the quotient

$$L_0^K / (L_{\text{loc}}^K \oplus \mathbb{Z}\mathbf{t}),\tag{14}$$

where $\mathbb{Z}\mathbf{t}$ is the sublattice consist of any integer multiple of $\mathbf{t}$. $L_{\text{loc}}^K \oplus \mathbb{Z}\mathbf{t}$ denotes the lattice generated by all the basis vectors of $L_{\text{loc}}^K$ and the vector $\mathbf{t}$. We readily observe that this is exactly the result of condensing the vison $\mathbf{t}$ in the original ungauged theory: $L_0^K / L_{\text{loc}}^K$ corresponds to all Abelian anyon types which braid trivially with $\mathbf{t}$. Further modding out $\mathbb{Z}\mathbf{t}$ identifies different anyon types that are related to each other by fusing with (multiples of) $\mathbf{t}$.

## 3 Gauging U(1) symmetry in a general MTC

We now proceed to describe U(1) gauging in a general gapped phase in (2+1)d. It is widely believed that bosonic gapped phases without any global symmetry in (2+1)d are completely classified by $(\mathcal{C}, c_-)$, where $\mathcal{C}$ is an MTC which encodes all universal properties of anyonic quasiparticles in the bulk, and $c_-$ is the chiral central charge of the edge. For a review of MTC in the context of (2+1)d topological phase, we refer the readers to Ref. [7]. In the following,

we will use the terms "quasiparticles" and "anyons" interchangeably. We will denote anyon types by $a, b, c, \ldots$, and the trivial anyon type (i.e. all local excitations) by 1.

To perform gauging, we need to first review how the global U(1) symmetry acts on the topological phase [11]. We will follow the general theoretical framework established in Ref. [1]. Without loss of generality, we can assume that the fundamental physical charge of the U(1) symmetry is 1. In a U(1)-symmetric topological phase, each quasiparticle $x$ carries a U(1) charge $Q_x$. Note that only $Q_x$ mod 1 is determined by the topological anyon type of $x$, since we may attach local U(1)-charged excitations to change $Q_x$ by any integer. By definition, the trivial anyon type 1, representing all local bosonic excitations, have $Q_1 = 0$. Here, notice the fact that U(1) being a continuous connected group cannot permute the anyon labels which are intrinsically discrete. The U(1) symmetry action on the topological phase is fully characterized by the fractional charge of each anyon. The charges $Q_a$ of the anyons must satisfy

$$Q_a + Q_b \equiv Q_c \text{ mod } 1, \text{ if } N_{ab}^c > 0. \tag{15}$$

Here $N_{ab}^c$ is the fusion coefficient, i.e. the multiplicity of anyon type $c$ in the fusion $a \times b$. As shown in Ref. [1], there exists a unique Abelian anyon $v$ such that

$$e^{2\pi i Q_a} = M_{av}, \tag{16}$$

for all anyon types $a$, where $M_{av}$ is the braiding phase between $a$ and $v$. Physically, $v$ is the excitation created by $2\pi$ flux insertion in this topological phase. By a straightforward generalization of the celebrated Laughlin argument, the charge $Q_v$ is given by $Q_v = \sigma_H$ mod 1, where $\sigma_H$ is the dimensionless Hall conductance. Formally, the Hall response is captured by the effective action

$$S = \frac{\sigma_H}{4\pi} \int A dA, \tag{17}$$

where $A$ is the background U(1) gauge field. In addition, one can show that $\sigma_H$ and $v$ are related by $e^{i\pi\sigma_H} = \theta_v$ [12,13], so $v$ determines $\sigma_H$ up to an even integer. The ambiguity is exactly the Hall conductance of bosonic integer quantum Hall states [13–15]. To summarize, the U(1) symmetry enrichment is fully determined by the vison $v$ and the Hall conductance $\sigma_H$.

We define $s$ to be the minimal positive integer such that $v^s = 1$, namely a minimal of $s$ copies of the anyon $v$ can fuse into the trivial anyon 1. It immediately follows that all fractional charges of the anyons are integer multiples of $1/s$. We further prove that there must exist a anyon carrying charge $1/s$, i.e. the minimal charge among the anyons is $e^* = 1/s$. Since $v^s$ is trivial, it also implies that $s^2\sigma_H$ must be an even integer because the topological twist factor $\theta_{v^s} = e^{i\pi s^2 \sigma_H}$ must be 1 for a trivial anyon. Also, note that $s\sigma_H$, being the U(1) charge carried by $v^s$, must be an integer.

Just like in the Abelian case, we need to treat $\sigma_H \neq 0$ and $\sigma_H = 0$ separately. First we assume $\sigma_H \neq 0$. Gauging the U(1) symmetry means that the background U(1) gauge field $A$ is promoted to a dynamical field. The Hall conductance implies that flux of $A$ must be attached to $A$ charge. In other words, the gauged theory only allow states that satisfy the constraint $\sigma_H \Phi + Q = 0$, which can be obtained from the equation of motion of Eq. (17). Here, $\Phi$ is the flux of the gauge field $A$. Thus a quasiparticle with charge $Q$ is attached a flux $-\frac{2\pi Q}{\sigma_H}$. This flux attachment changes the topological twist factor by a factor of $\exp\left(-i\pi\frac{Q^2}{\sigma_H}\right)$ via the Aharonov-Bohm effect between the flux and the charge of $A$. Based on these observations, we now describe the anyon content of the gauged system.

First of all, due to flux attachment it is sufficient to keep track of the charge of an excitation. In the ungauged theory, the anyon type $a \in \mathcal{C}$ only determines the charge $Q_a$ mod 1, and excitations belonging to the same anyon type $a$ can have any value of charge $Q_a + n$ where

$n \in \mathbb{Z}$. After gauging, naively all these different charges become distinct superselection sectors. Therefore, it will prove to be convenient to first enlarge our set of anyons by including explicitly the charge quantum number. Namely, we now consider all possible excitations $(a, Q_a)$, subject to the constraint $e^{2\pi i Q_a} = M_{av}$. Formally we are working with a much larger theory with (countably) infinite many types of particles [2] We will denote this intermediate category by $\mathcal{C}'$.

One reason to take this detour is that the topological data of this enlarged category $\mathcal{C}'$ can be explicitly written down, which we present now. First of all, the identity object is $(1, 0)$, so the anti-particle of $(a, Q_a)$ is $(\bar{a}, -Q_a)$, where $\bar{a}$ denotes the anti-particle of the anyon $a$ within the MTC $\mathcal{C}$. The fusion rules are given by

$$(a, Q_a) \times (b, Q_b) = \sum_c N_{ab}^c (c, Q_a + Q_b). \tag{18}$$

The flux attachment also modifies braiding and exchange statistics between anyons, by additional phase factors from Aharonov-Bohm effect between charge and flux. They can be computed solely from the effective CS response. The $S$-matrix of the intermediate category $\mathcal{C}'$ is hence given by

$$S_{(a,Q_a),(b,Q_b)} = S_{ab} e^{2\pi i \frac{Q_a Q_b}{\sigma_H}}, \tag{19}$$

with $S_{ab}$ the topological $S$-matrix of the ungauged theory $\mathcal{C}$. The topological twist factors in the category $\mathcal{C}'$ become

$$\theta_{(a,Q_a)} = \theta_a e^{-\pi i \frac{Q_a^2}{\sigma_H}}, \tag{20}$$

where $\theta_a$ is the topological twist factor of the anyon $a$ in the ungauged theory. It is straightforward to check that the $S$-matrix, topological twist factors and fusion rules satisfy compatibility conditions expected from axioms of braided fusion categories, even though the number of anyon types is now infinity.

We argue that the flux attachment induced by the dynamics of $A$ does not lead to non-trivial contributions to the $F$-symbols of $\mathcal{C}'$, since the exact charges are fused. Therefore, we expect the following $F$- and $R$-symbols for this enlarged infinite category $\mathcal{C}'$:

$$[F_{(d,Q_d)}^{(a,Q_a),(b,Q_b),(c,Q_c)}]_{(e,Q_e),(f,Q_f)} = [F_d^{abc}]_{ef},$$
$$R_{(c,Q_c)}^{(a,Q_a),(b,Q_b)} = R_c^{ab} e^{-\frac{\pi i}{\sigma_H} Q_a Q_b}. \tag{21}$$

One can readily show that pentagon and hexagon equations are satisfied, and the data correctly reproduce the topological $S$-matrix given in Eq. (19) and the topological twist factors given in Eq. (20).

Next, we truncate the infinite category $\mathcal{C}'$ to obtain a more physical description of the gauged theory. We expect that after gauging, the system remains gapped (because of a non-zero Hall conductance), so should be described by a new MTC $\mathcal{D}$. This gauged theory $\mathcal{D}$ will have different $F$- and $R$-symbols compared to those of $\mathcal{C}'$ given in Eq. (21) because of the truncation. We begin with a formal approach: first identify the transparent anyons of the enlarged infinite category $\mathcal{C}'$. Here, transparent anyons are those anyons that have trivial braiding with every other anyon, so they should be understood as local excitations (of the gauged system). Suppose $(a, Q_a)$ braids trivially with all anyons in $\mathcal{C}'$. The anyon $a$ should be Abelian. Then

$$M_{(a,Q),(b,Q_b)} = M_{ab} e^{-2\pi i \frac{Q_a Q_b}{\sigma_H}} = 1. \tag{22}$$

So we must have $M_{ab} = e^{2\pi i \frac{Q_a}{\sigma_H} Q_b}$ for all $b$ and $Q_b$, and apparently the relation is invariant under $Q_b \to Q_b + 1$. Thus we find $e^{2\pi i \frac{Q_a}{\sigma_H}} = 1$, or $Q_a = n\sigma_H$. Because $M_{ab} = e^{2\pi i n Q_b} = M_{v^n, b}$

---

[2]Technically speaking, this enlarged category should satisfy all axioms of ribbon fusion category except the finiteness condition.

for all $b$, by modularity of $\mathcal{C}$ we must have $a = v^n$. Thus the group of transparent anyons in $\mathcal{C}'$ is generated via fusion by the anyon $(v, \sigma_H)$ whose self-statistics is bosonic, namely $\theta_{(v,\sigma_H)} = 1$. Denote $T_k = (v^k, k\sigma_H)$ for $k \in \mathbb{Z}$ and note that $T_k = T_{k'}$ if and only if $k = k'$ (because of a non-zero Hall conductance $\sigma_H$). We thus condense the group of transparent anyons, generated by $T_1$, and obtain a new MTC $\mathcal{D}$, which describes the topological order resulting from gauging the U(1) symmetry of the original theory $\mathcal{C}$. It is easy to see that after the condensation the theory contains finitely many anyons. Observe that $T_s = (1, s\sigma_H)$ is condensed. Therefore the charges in the category can be restricted to the range $0 \leq Q_a < s\sigma_H$. Thus for each $a$, there are only a finite number of choices for $Q_a$ that need to be considered. In fact, with proper conventions, one can even restrict $Q_a$ to range $0 \leq Q_a < \sigma_H$ in the gauged theory $\mathcal{D}$ as we can see in the examples later.

Physically, it is quite evident that the transparent particle $(v, \sigma_H)$ should be interpreted as the vison $v$ but with $-2\pi$ flux of $A$ attached. Since $v$ is created by the $2\pi$ flux insertion, the composite $(v, \sigma_H)$ becomes a local excitation and therefore must be condensed.

Mathematically, the advantage of going through the intermediate steps (i.e. enlarging the category and then condensing transparent anyons) is that the approach allows us to write down the full topological data, especially the $F$- and $R$-symbols, of the resulting MTC $\mathcal{D}$ explicitly, in terms of those of the original MTC, and the U(1) symmetry enrichment data $v$ and $\sigma_H$. The details of the procedure are given in App. B.

While the general expressions of the $F$- and $R$-symbols of the MTC $\mathcal{D}$ that describes the gauged theory are rather complicated, the topological $S$-matrix and the topological twist factors of $\mathcal{D}$ can be simply written down in terms of those of the original MTC $\mathcal{C}$. As discussed above, the MTC $\mathcal{D}$ can be obtained from condensing $(v, \sigma_H)$ in the infinite category $\mathcal{C}'$. When $(v, \sigma_H)$ is condensed, the anyons $(a, Q_a) \in \mathcal{C}'$ that are only different from each other by the fusion with an integer copies of $(v, \sigma_H)$ are identified as a single type of anyon in $\mathcal{D}$. In other words, the anyons in $\mathcal{C}'$ form orbits under the fusion with $(v^k, k\sigma_H)$ for $k \in \mathbb{Z}$. Each different orbit corresponds to a different type of anyon in $\mathcal{D}$. The topological twist factor of an anyon in $\mathcal{D}$ is the same as that of any representative anyon within the corresponding orbit in $\mathcal{C}'$, which can be calculated using Eq. (20). Since one can easily show that an orbit of anyons in $\mathcal{C}'$ shares the same topological twist factor $\theta_{(a,Q_a)}$, the choice of representative does not affect the result. The topological $S$-matrix of the MTC $\mathcal{D}$ can be obtained in a similar fashion. The $S$-matrix element between two anyons of $\mathcal{D}$ is the same as the $S$-matrix element between their respective representatives in $\mathcal{C}'$ which is given in Eq. (19). An immediate consequence is that an anyon in $\mathcal{D}$ has the same quantum dimension as its representative in $\mathcal{C}'$.

So far we have treated the category $\mathcal{C}'$ as a pure mathematical device. One may also wonder whether $\mathcal{C}'$ has any physical meaning. In App. F, we provide a "holographic" viewpoint on gauging, where $\mathcal{C}'$ appears as a particular boundary theory for a (3+1)d Maxwell U(1) gauge theory. Going from $\mathcal{C}'$ to $\mathcal{D}$ then corresponds to the confinement transition in the (3+1)d bulk.

At this point, the only missing information about the gauged system is the chiral central charge $c'_-$. Since we have determined the bulk anyon data of the gauged system, we can evaluate $c'_-$ mod 8, by using the generalized Gauss-Milgram sum. As shown in App. D, we find

$$c'_- = c_- - \operatorname{sgn}\sigma_H \mod 8. \tag{23}$$

In fact, we believe that $c'_- = c_- - \operatorname{sgn}\sigma_H$ holds exactly. In the Abelian case, the relation was proven by the field theory construction. Now that we've obtained all the data of the gauged theory, our gauging procedure is completed. Note that gauging a finite group leaves the chiral central charge unchanged [1], which is fundamentally different from the case when a U(1) symmetry is gauged.

Moreover, it is interesting to observe that the gauged theory $\mathcal{D}$ is mathematically equivalent

to the following MTC

$$\mathcal{C} \boxtimes U(1)_{-s^2\sigma_H}\big|_{(v,s\sigma_H)}, \tag{24}$$

where we stack the MTC $\mathcal{C}$ with an additional layer of MTC described by the $U(1)_{-s^2\sigma_H}$ CS theory, and condense the composite anyon $(v, s\sigma_H) \in \mathcal{C} \boxtimes U(1)_{-s^2\sigma_H}$ with $v \in \mathcal{C}$ and $s\sigma_H \in U(1)_{-s^2\sigma_H}$. Such a MTC $\mathcal{C} \boxtimes U(1)_{-s^2\sigma_H}\big|_{(v,s\sigma_H)}$ can also be viewed as a "hierarchy construction" proposed in Ref. [16] which is a categorical formulation of the hierarchy construction of the fractional quantum Hall states [17–19]. In the hierarchy construction, the additional layer $U(1)_{-s^2\sigma_H}$ is also a U(1) SET, with $s\sigma_H$ being the vison. Then after condensing the bound state of visons from both layers, the remaining anyons must be the composite of an anyon $a \in \mathcal{C}$ and an anyon with a gauge charge $q$ in the U(1) CS theory satisfying the condition $M_{av} = e^{2\pi i \frac{q}{s}}$. Note that in the Abelian layer the anyon $q$ carries U(1) charge $Q_q = -q/s$, so by pairing up $a$ in $\mathcal{C}$ with $q$ in $U(1)_{-s^2\sigma_H}$, after condensation all remaining anyons are charge-neutral. Another useful observation is that the subcategory of $\mathcal{C} \boxtimes U(1)_{-s^2\sigma_H}$ that braids trivially with $(v, s\sigma_H)$ can already be identified with the premodular category $\mathcal{D}_{\text{int}}$ which is obtained in App. B.2 as an intermediate step towards the full U(1) gauging of $\mathcal{C}$. With this observation, it is straightforward to check this hierarchy construction indeed gives the MTC $\mathcal{D}$ which is the outcome of gauging the U(1) symmetry of $\mathcal{C}$. This result Eq. 24 can be also understood using a gauging procedure based symmetry group extensions [20].

Below we provide an example of gauging a U(1) symmetry in $SU(2)_k$ MTC, and as a result obtaining all categorical data for $\mathbb{Z}_k$ parafermion MTCs. In the next section, we will discuss the gauging of the U(1) symmetry when the Hall conductance vanishes, i.e. $\sigma_H = 0$.

### 3.1 Example: Gauging U(1) SPT phase

Consider the simplest case $\mathcal{C} = \text{Vec}$, i.e. a U(1) bosonic SPT phase, with Hall conductance $\sigma_H = n \in 2\mathbb{Z}$. The only "anyon" in the MTC $\mathcal{C} = \text{Vec}$ is the trivial one, denoted as "1" in the following. In this case, the vison $v$ has to be the trivial anyon, namely $v = 1$. Let us assume a positive Hall conductance, i.e. $\sigma_H = n > 0$ in the following. The intermediate infinite category $\mathcal{C}'$ consists of all charges $\{(1, q)\}$ for $q \in \mathbb{Z}$ (where the first entry "1" denote the trivial anyon in $\mathcal{C} = \text{Vec}$). Condensing $(v, \sigma_H) = (1, n)$ in $\mathcal{C}'$, we are left with $n$ different anyons $(1, a)$ for $0 \le a < n$ and $a \in \mathbb{Z}$ whose the fusion rule is given by

$$(1, a) \times (1, b) = (1, \lceil a + b \rfloor_n), \tag{25}$$

where $\lceil a + b \rfloor_n$ is the residue of $a + b$ modulo $n$. Using Eq. (36) we find that the $F$-symbol of the gauged theory $\mathcal{D}$ is given by

$$F^{abc} = e^{-\frac{\pi i}{n}a(b+c-\lceil b+c \rfloor_n)}, \tag{26}$$

and $R$- symbol is given by

$$R^{ab} = e^{-\frac{\pi i}{n}ab}. \tag{27}$$

The resulting MTC $\mathcal{D}$ describes an Abelian topological order. In the $F$- and $R$-symbols given above, the suppressed indices are completely determined via the fusion rule by the indices that are explicitly written. The resulting MTC data agrees with what is known as the $U(1)_{-n}$, or $\mathbb{Z}_n^{(-1/2)}$ MTC. One can easily confirm this result using a field-theoretic approach. Before gauging the U(1) symmetry, the U(1) bosonic SPT phase simply produces a non-trivial response theory with an action $S = \frac{n}{4\pi} \int A dA$ for the background $A$ gauge field. The constituent matter fields of the U(1) bosonic SPT phase have already been safely integrated out here because the U(1) bosonic SPT state only carries invertible topological order. When the U(1) symmetry is gauged, $A$ becomes a dynamical U(1) gauge field. The same action $\frac{n}{4\pi} \int A dA$ now describes the (2+1)d topological order of $U(1)_{-n}$.

## 3.2 Example: $\mathbb{Z}_k$ parafermion

Consider a (2+1)d topological phase described by the SU(2)$_k$ MTC, namely $\mathcal{C} = \text{SU}(2)_k$. There are $k + 1$ different types of anyons, labeled by $j = 0, 1/2, \cdots, k/2$. Note that there is a single nontrivial Abelian anyon $k/2$.

This system naturally admits a SO(3) global symmetry: Anyons of type $j$ carry spin-$j$ representation of SO(3). We now consider gauging a U(1) subgroup of SO(3), with the vison associated with the $2\pi$ flux given by $v = k/2$. Since $\theta_{k/2} = e^{\pi i k/2}$, the Hall conductance is $\sigma_H = \frac{k}{2}$ mod $2\mathbb{Z}$. While the formalism is general, the even $k$ case is of most interest. When $k$ is odd, the anyon $k/2$ is a semion and the whole MTC factorizes into a "SO(3)$_k$" MTC and the semion theory. The U(1) symmetry only acts non-trivially on the semion part. So in the following we assume $k$ is even.

In App. B.3, we carry out the gauging procedure in full detail and derive the complete set of topological data for the gauged theory $\mathcal{D}$. When $\sigma_H = \frac{k}{2}$, we show that the resulting MTC the $\mathbb{Z}_k$ parafermion MTC. The parafermion MTC, initially derived from the $\mathbb{Z}_k$ parafermion conformal field theory introduced by Ref. [21], is closely related to the topological order of the Read-Rezayi fractional quantum Hall states [22]. In the following, we will study how the $\mathbb{Z}_k$ parafermion MTC arises from U(1) gauging in SU(2)$_k$ using a field-theoretic formalism, to corroborate the algebraic approach.

Before gauging, we can describe the U(1) symmetric SU(2)$_k$ MTC using a dynamical SU(2)$_k$ Chern-Simons theory. We denote the gauge connection of the SU(2) gauge field as $a = \frac{1}{2}\sigma^1 a^1 + \frac{1}{2}\sigma^2 a^2 + \frac{1}{2}\sigma^3 a^3$ where the Pauli matrices $\frac{1}{2}\sigma^{1,2,3}$ are the generators of the SU(2) gauge group and the 1-forms $a^{1,2,3}$ are the three associated components of the SU(2) gauge field. Let the 1-form $A$ denote the background gauge field associated with the U(1) symmetry.

The anyon $j$ in the SU(2)$_k$ MTC corresponds to the spin-$j$ representation of the SU(2) gauge group. A natural way to assign $j/2$ charge to the anyon $j$ for all $j = 0, 1, 2, ..., k/2$ is to consider embedding the U(1) and SU(2) gauge connections together into a single U(2) gauge field $b$:

$$b = \frac{1}{2}\mathbb{1}A + \frac{1}{2}\sigma^1 a^1 + \frac{1}{2}\sigma^2 a^2 + \frac{1}{2}\sigma^3 a^3, \tag{28}$$

where $\mathbb{1}$ is the $2 \times 2$ identity matrix. The $j = 1/2$ anyon which carries the two-dimensional representation under the U(2) gauge group naturally carries U(1) charge 1/2. More generally, an anyon $j$ should carry a representation under the gauge group U(2) that is given by the symmetric combination of $2j$ copies of the representation carried by the anyon labeled by $j = 1/2$. Therefore, the anyon $j$ carries U(1) charge $j/2$. Now, we consider the following Lagrangian that decribes a SU(2)$_k$ topological order under a background U(1) gauge field:

$$\mathcal{L} = -\frac{k}{4\pi}\text{Tr}\left(bdb - \frac{2\text{i}}{3}b^3\right) + \frac{k}{4\pi}(\text{Tr}\, b)d(\text{Tr}\, b), \tag{29}$$

which is also known as the U(2)$_{k,-k}$ Chern-Simons term [23, 24]. Our convention for the Lagrangian of the U(2)$_{k,-k}$ Chern-Simons term contains an extra overall minus sign compared to Ref. [24]. This extra minus sign is due to our convention of the sign of the chirality of the corresponding MTC with respect to the sign of the Hall conductance. When we turn off the background U(1) gauge field by setting $A = 0$, the Lagrangian $\mathcal{L}$ above is reduced to $-\frac{k}{4\pi}\text{Tr}\left(ada - \frac{2\text{i}}{3}a^3\right)$ which describes the SU(2)$_k$ topological order. With a finite background U(1) gauge field $A$, this Lagrangian $\mathcal{L}$ effectively contains an extra term "$\frac{k}{8\pi}AdA$" almost decoupled from the SU(2)$_k$ part, which indicates a Hall conductance of $\sigma_H = k/2$. Here, strictly speaking, "$\frac{k}{8\pi}AdA$" is not truly well-defined on its own and the U(1) background gauge field can not be fully decoupled from the dynamical SU(2) gauge field $a$: the embedding of U(1) and SU(2) gauge fields into a single U(2) gauge field requires an identification between the

$\mathbb{Z}_2$ subgroup of U(1) with the center $\mathbb{Z}_2$ of SU(2). Upon gauging the U(1) symmetry of the $SU(2)_k$ topological order, the theory simply becomes a fully dynamical $U(2)_{k,-k}$ Chern-Simons gauge theory. Following Ref. [24], the TQFT can be identified as

$$U(2)_{k,-k} \equiv \frac{SU(2)_k \times U(1)_{-2k}}{\mathbb{Z}_2}, \tag{30}$$

where the quotient $\mathbb{Z}_2$ means that the anyon $(k/2, k) \in SU(2)_k \times U(1)_{-2k}$ in $SU(2)_k \times U(1)_{-2k}$. (The notation for the anyons in $SU(2)_k \times U(1)_{-2k}$ will be explained in detail in the following). We recognize that the right-hand side of Eq. (30) is exactly the hierarchy construction for $\mathcal{D}$ in Eq. (24). Below we will explicitly check the equivalence between the hierarchy construction and the gauging procedure as described in Sec. 3 (and App. B).

Let's first unpack the algebraic definition of $U(2)_{k,-k}$. One can start with the MTC given by $\mathcal{B}_0 = SU(2)_k \times U(1)_{-2k}$ whose anyons are labeled by the pair $(j, n)$. Here, $j = 0, 1, ..., k/2$ labels the anyon type within the $SU(2)_k$ sector and $n = 0, 1, ..., 2k$ labels the anyon type within the $U(1)_{-2k}$ sector. The fusion rule of the Abelian topological order $U(1)_{-2k}$ is given by $n_1 \times n_2 = \lceil n_1 + n_2 \rceil_{2k}$, where $\lceil \cdot \rceil_{2k}$ denotes the residue of $\cdot$ modulo $2k$.

To obtain the $\mathbb{Z}_k$ parafermion MTC, one needs to condense the anyon $(j = k/2, n = k)$ in $SU(2)_k \times U(1)_{-2k}$. Note that the anyon $(k/2, k)$ has a bosonic self-statistics but is not a transparent anyon in $\mathcal{B}_0$. One can define a sub-category $\mathcal{B}_1 \subset \mathcal{B}_0$ consists of only anyons in $\mathcal{B}_0$ that braid trivially with the anyon $(k/2, k)$. The anyons in the sub-category $\mathcal{B}_1$ are the anyons $(j, n)$ such that $2j + n \in 2\mathbb{Z}$. The $F$- and $R$-symbols of category $\mathcal{B}_1$ are given by

$$\left( F^{(j_1, n_1),(j_2, n_2),(j_3, n_3)}_{(j, n)} \right)_{(j_{12}, n_{12}),(j_{23}, n_{23})} = \left( F^{j_1, j_2, j_3}_j \right)_{j_{12}, j_{23}} e^{-i\frac{\pi}{2k} n_1(n_2 + n_3 - \lceil n_2 + n_3 \rceil_{2k})},$$
$$R^{(j_1, n_1),(j_2, n_2)}_{(j, n = \lceil n_1 + n_2 \rceil_{2k})} = R^{j_1, j_2}_j e^{-i\frac{2\pi}{4k} n_1 n_2}. \tag{31}$$

Each of the $F$- and $R$-symbols of the category $\mathcal{B}_1$ in Eq. (31) is a product of two factors. $\left( F^{j_1, j_2, j_3}_j \right)_{j_{12}, j_{23}}$ and $R^{j_1, j_2}_j$ are the $F$- and $R$-symbols of $SU(2)_k$ while the factors $e^{-i\frac{\pi}{2k} n_1(n_2 + n_3 - \lceil n_2 + n_3 \rceil_{2k})}$ and $e^{-i\frac{2\pi}{4k} n_1 n_2}$ are the contributions from the $U(1)_{-2k}$ sector. By definition, the category $\mathcal{B}_1$ is a premodular category with the anyon $(k/2, k)$ the transparent Abelian anyon. By condensing $(k/2, k)$ in $\mathcal{B}_1$, we can obtain the $\mathbb{Z}_k$ parafermion MTC.

Interestingly, before we condense the anyon $(k/2, k) \in \mathcal{B}_1$, the premodular category $\mathcal{B}_1$ can already be identified as a category $\mathcal{D}_{\text{int}}$ obtained as an intermediate step of gauging the U(1) symmetry of $SU(2)_k$. The procedure of gauging the U(1) symmetry of $SU(2)_k$ starts with considering the category $\mathcal{C}'$ with infinitely many anyons $(j, Q)$ defined by the data in Eq. (18) and in Eq. (21). Gauging the U(1) symmetry amounts to condensing the anyon $(\nu, \sigma_H = k/2) = (j = k/2, Q = k/2) \in \mathcal{C}'$. As explained in App. B.2, we can first condense the anyon $(1, s\sigma_H)$ in $\mathcal{C}'$ to obtain an intermediate finite category, denoted as $\mathcal{D}_{\text{int}}$, with $s = 2$ due to the $\mathbb{Z}_2$ fusion rule of the anyon $j = k/2$ in $SU(2)_k$. The $F$- and $R$-symbols of the category $\mathcal{D}_{\text{int}}$ can be calculated using Eq. (40) and Eq. (41). One can see that the so-obtained $F$- and $R$-symbols of the category $\mathcal{D}_{\text{int}}$ are the same as the those of the category $\mathcal{B}_1$ shown in Eq. (31) upon identifying the anyon $(j, Q)$ of $\mathcal{D}_{\text{int}}$ as the anyon $(j, n = 2Q)$ of $\mathcal{B}_1$. Therefore, the categories $\mathcal{D}_{\text{int}}$ and $\mathcal{B}_1$ are completely identical. By further condensing the transparent anyon $(\nu, \sigma_H = k/2) = (j = k/2, Q = k/2)$ in $\mathcal{D}_{\text{int}}$ (or equivalently the transparent anyon $(j = k/2, k)$ in $\mathcal{B}_1$), we can complete the procedures for gauging the U(1) symmetry of the $SU(2)_k$ MTC and, as the result, obtain the $\mathbb{Z}_k$ parafermion MTC.

The simplest example is the case of $k = 2$, namely the $SU(2)_2$ MTC. When the Hall conductance is $\sigma_H = k/2 = 1$, gauging the U(1) symmetry results in the $\mathbb{Z}_2$ parafermion MTC which is more commonly referred to as the Ising MTC. Interestingly, when we change Hall conductance to $\sigma_H = -1$, the resulting MTC becomes $Spin(5)_1$ instead, which is a close relative to the Ising MTC.

It is not difficult to generalize the discussion to gauging a U(1) symmetry in $SU(N)_k$, which results in a $U(N)_{k,k+Nk'}$ MTC (when $k + Nk' \neq 0$ and $k + k'$ even). More details can be found in App. E.

# 4 Gauging U(1) symmetry when $\sigma_H = 0$

We now discuss what happens when $\sigma_H$ vanishes. In this case, the vison $v$ must have a bosonic self-statistics, i.e. $\theta_v = e^{i\pi\sigma_H} = 1$. We have shown in the Abelian case in Sec. 2 that gauging the U(1) symmetry has the same effect as condensing $v$ in the original theory $\mathcal{C}$, and results in a gauged theory with U(1)-charge-neutral excitations. In App. E, we show, using a combination of field-theoretic and algebraic approaches, that the same is also true when we gauge the U(1) symmetry of the $SU(N)_k$ MTC with a vanishing Hall conductance. Physically, a vanishing Hall conductance $\sigma_H = 0$ leads to the absence of a CS term for the U(1) gauge field $A$. When the gauge field $A$ becomes dynamical in the absence of any CS term, the instantons of the gauge field $A$ are expected to proliferate resulting in the condensation of $2\pi$ flux which is tied to the vison $v \in \mathcal{C}$. Therefore, we expect that, for a general U(1)-symmetric (2+1)d topological order described by the MTC $\mathcal{C}$, gauging the U(1) symmetry of $\mathcal{C}$ has the same effect as the condensing the vison $v$ in MTC $\mathcal{C}$. Such condensation is permissible because of $\theta_v = 1$ (a consequence of the vanishing Hall conductance) and yields a new MTC $\mathcal{D}$ as the result of gauging the U(1) symmetry of original theory $\mathcal{C}$.

Since condensing an Abelian anyon with a bosonic self-statistics does not change the chiral central charge of the MTC, gauging the U(1) symmetry of (2+1)d topological order with a vanishing Hall conductance leaves the chiral central charge invariant.

# 5 Dual $U(1)_{\text{dual}}$ symmetry after U(1) gauging

When gauging a finite group $G$, the gauged theory always has a subcategory isomorphic to $\text{Rep}(G)$ (i.e. the symmetric tensor category of irreducible linear representations of $G$). Condensing $\text{Rep}(G)$ performs "ungauging" and returns to the original theory [1–5]. When $G$ is a finite Abelian group, this phenomenon can be equivalently formulated as the gauged theory having a (non-anomalous) 1-form symmetry group isomorphic to $G$, and gauging the 1-form symmetry produces the ungauged theory [25, 26]. In other words, gauging a finite Abelian (0-form) symmetry leads to a dual 1-form symmetry. Notice that this statement holds independent of the actual low-energy dynamics of the theory.

We now describe the analogy of dual symmetry in the case of U(1) gauging. Let's consider a U(1)-symmetric (2+1)d topological order described by the MTC $\mathcal{C}$. We start by assuming a non-vanishing Hall conductance, i.e. $\sigma_H \neq 0$. After gauging the U(1) symmetry of $\mathcal{C}$, we obtain a new topological order whose corresponding MTC is $\mathcal{D}$. The procedure in obtaining the categorical data of the MTC $\mathcal{D}$ is discussed in Sec. 3. In fact, the topological order $\mathcal{D}$ can admit a dual 0-form U(1) symmetry, which we denote as $U(1)_{\text{dual}}$. From the field theory perspective, the current of this $U(1)_{\text{dual}}$ symmetry is given by $\star dA/(2\pi)$ with $A$ the gauge field associated with the original U(1) symmetry (before it is gauged) and $\star$ the Hodge star operator. Physically, the $U(1)_{\text{dual}}$ symmetry is the conservation of magnetic flux of the original U(1) symmetry. When the original U(1) symmetry is gauged, $dA/(2\pi)$ becomes a dynamical object which can be minimally coupled to the (non-dynamical) background $U(1)_{\text{dual}}$ gauge field $A^{\text{dual}}$ via

$$\int \frac{1}{2\pi} A^{\text{dual}} dA. \tag{32}$$

Since the dynamics of $A$ is effectively governed by the action Eq. (17), the Hall conductance of the gauged theory $\mathcal{D}$ with respect to the $U(1)_{\text{dual}}$ symmetry is given by

$$\sigma'_H = -1/\sigma_H \,. \tag{33}$$

There is a new vison $v' \in \mathcal{D}$ in the gauged theory $\mathcal{D}$ that is associated with the $2\pi$ flux insertion of the $A^{\text{dual}}$. Eq. (32) implies that the vison $v'$ should be identified just as the unit charge of $A$, which further corresponds to the anyon $(1, 1)$ in the infinite category $\mathcal{C}'$ introduced in Sec. 3. As discussed above, the gauged theory $\mathcal{D}$ can be obtained from the infinite category $\mathcal{C}'$ via the condensation of the transparent anyon $(v, \sigma_H) \in \mathcal{C}'$. Since this condensation does not change the topological twist factors, we can obtain the topological twist factor of the vison $v' \in \mathcal{D}$ using Eq. (20):

$$\theta_{v'} = \theta_{(1,1)} = e^{-i\pi/\sigma_H} \,. \tag{34}$$

Notice that the consistency condition $\theta_{v'} = e^{i\pi\sigma'_H}$ is satisfied in the gauged theory $\mathcal{D}$ with the $U(1)_{\text{dual}}$ symmetry. If one consider further gauging the $U(1)_{\text{dual}}$ symmetry of the theory $\mathcal{D}$, it is obvious from the field theory perspective that resulting theory should be identical to the original theory $\mathcal{C}$ with the same $U(1)$ symmetry enrichment, namely the same Hall conductance $\sigma_H$ and same vison $v \in \mathcal{C}$, as before.

It can happen that in going from $\mathcal{C}'$ to $\mathcal{D}$, $(1, 1)$ is also condensed when it is generated by $(v, \sigma_H) \equiv T_1$. In other words, if this happens we must have $(1, 1) = (v^s, s\sigma_H)$, which implies $\sigma_H = \frac{1}{s}$. In this case, since $\theta_v = e^{i\pi/s}$, the subcategory generated by $v$ is identified with $U(1)_s$, which is an MTC on its own. By the factorization property of MTCs, it implies that the original MTC $\mathcal{C}$ takes the form $\mathcal{C} = \mathcal{D} \boxtimes U(1)_s$, and it is not difficult to see that $\mathcal{D}$ is indeed the gauged theory (hence the notation). In this case the dual vison is $v' = 1$.

When $\sigma_H = 0$, the most relevant term of the gauge field $A$ generated by the "matter fields" in the original theory $\mathcal{C}$ is a (2+1)d Maxwell term. When the original $U(1)$ symmetry is gauged, the gauge field $A$ is governed by a Maxwell theory, whose flux is now conserved because of the $U(1)_{\text{dual}}$ symmetry. The Polyakov's instanton proliferation mechanism is forbidden by the $U(1)_{\text{dual}}$ symmetry. In this case, assuming that the matter fields of the original theory $\mathcal{C}$ remains gapped, the resulting phase is a gapless phase whose low-energy modes are given by the deconfined and gapless photons of the gauge field $A$. In fact, in this phase, the $U(1)_{\text{dual}}$ symmetry is spontaneously broken and the corresponding Goldstone modes are dual to the gapless photons of the gauge field $A$. In this gapless phase, it is no longer appropriate to characterize the resulting phase of matter using just MTCs. Note that if there is no $U(1)_{\text{dual}}$ symmetry presence, the confinement of the gauge field $A$ through the instanton proliferation mechanism should always happen resulting in a gapped phase as the gauged theory. And the resulting topological order follows from the discussion of Sec. 4.

In principle, one can also consider the scenario when the gauge field $A$ becomes higgsed. In this scenario, $U(1)_{\text{dual}}$ symmetry is no longer spontaneously broken and the resulting phase should be gapped. The topological order of this $U(1)_{\text{dual}}$-symmetric gapped phase will depend on the choice of the Higgs field. The most trivial situation is when the Higgs field is topologically equivalent to the trivial anyon in the original theory $\mathcal{C}$ before the original $U(1)$ symmetry is gauged. The resulting topological order is still given by an MTC $\mathcal{C}$. But its enrichment under the $U(1)_{\text{dual}}$ symmetry is trivial.

# 6 Discussion

MTCs from Chern-Simons theory of compact Lie groups are closely related to Wess-Zumino-Witten (WZW) chiral conformal field theories in (1+1)d. It is known that gauging a subgroup

in a WZW theory is equivalent to the coset construction of the WZW CFT [27–29]. For example, $\mathbb{Z}_k$ parafermion CFT can be viewed as the coset $\mathrm{SU}(2)_k/\mathrm{U}(1)$ theory. Thus the gauging prescription we gave is basically the categorical version of coset by U(1). Since coset construction works for any Lie group symmetry, it is an important question to develop a categorical description of gauging for general compact Lie groups. We address this question in a follow-up publication [20]

In the finite group case, Ref. [1] describes the gauging in two steps: first symmetry defects are introduced and together with anyons they form a mathematical structure called $G$-crossed braided tensor category. Then an "equivariantization" procedure is applied to obtained an MTC corresponding to the gauged system, which is physically the projection to $G$-invariant subspace. Formally one can also define "U(1)-crossed" braided category, as shown in Ref. [1] and Ref. [11, 30, 31], where symmetry defects are labeled by elements of U(1). It will be interesting to understand how this approach to gauging is related to ours.

Another direction for future work is to generalize the construction to fermionic systems with $\mathrm{U}(1)_f$ symmetry, where $\mathrm{U}(1)_f$ is the conservation of fermion number. In other words, local excitations with odd/even charge are fermions/bosons. We expect that the basic strategy in this work can be generalized, but there may be additional sign factors in the $F$- and $R$-symbols for the category $\mathcal{C}'$ coming from Fermi statistics.

Moreover, a recent work Ref. [32] provides a general analysis on the coupling between (2+1)d topological orders and general curved U(1) background gauge fields. Establishing the relation between the analysis in Ref. [32] and our general framework for U(1) symmetry gauging (where the U(1) gauge field becomes dynamical) will be left for future work.

## Acknowledgments

We thank M. Barkeshli for enlightening conversations and sharing unpublished results, and P. Bonderson and T. Lan for feedbacks on a draft of the manuscript. We are especially grateful to P. Bonderson for explaining the hierarchy construction for fractional quantum Hall states. C.-M.J. thanks D. Aasen for helpful discussions. M.C. acknowledges support from NSF under award number DMR-1846109.

## A   Minimal charge

The excitation $v$ induced by a $2\pi$ flux must be an Abelian anyon. Suppose $v$ has $\mathbb{Z}_s$ fusion rule in the bosonic MTC (before gauging U(1)), let's show that the minimal charge $e^*$ has to be $\frac{1}{s}$. For any anyon $a$, since $e^{2\pi i Q_a} = M_{av}$, the $\mathbb{Z}_s$ fusion rule of $v$ requires $sQ_a \in \mathbb{Z}$. All possible charges that can by any anyonic or local excitation in this MTC is given by $(\oplus_a Q_a \mathbb{Z}) \oplus \mathbb{Z}$. Our goal is to show that $\frac{1}{s} \in (\oplus_a Q_a \mathbb{Z}) \oplus \mathbb{Z}$.

Let's prove it by contraction. Let's assume $\frac{1}{s} \notin (\oplus_a Q_a \mathbb{Z}) \oplus \mathbb{Z}$. It implies that greatest common divisor $n$ of the set of integers $\{sQ_a\}_a \cup \{s\}$ is greater than 1, i.e. $n > 1$. Then, we can consider an non-trivial anyon $x$ which is the fusion product of $\frac{s}{n}$ anyons $v$. $x$ is an non-trivial anyon because $v$ has $\mathbb{Z}_s$ fusion rules. Notice that $M_{xa} = (M_{va})^{\frac{s}{n}} = e^{2\pi i Q_a s/n} = 1$ for all anyon $a$. However, in an MTC, the only "anyon" with this property is the trivial anyon. Hence, we arrive at a contradiction. And therefore, we prove that $\frac{1}{s} \in (\oplus_a Q_a \mathbb{Z}) \oplus \mathbb{Z}$, namely $e^* = \frac{1}{s}$. This statement further implies that $(0, \sigma_H/e^*)$ is always generated by $(v, Q_v)$.

# B  Condensing transparent anyons

## B.1  General Formalism

As an intermediate step in gauging the U(1) symmetry of the MTC $\mathcal{C}$, we have introduced in Sec. 3 the infinite category $\mathcal{C}'$ are contains all possible excitations $(a, Q_a)$. The fusion rule of this infinite category $\mathcal{C}'$ is given by Eq. (18) and the $F$- an $R$-symbols are given by Eq. (21). For simplicity, we will only focus on the situation where all of the fusion multiplicity $N_{ab}^c$ of the original MTC $\mathcal{C}$ before gauging U(1) is less than or equal to 1, i.e. $N_{ab}^c \leq 1$. In this appendix, we use a a single greek letter to denote the pair $(a, Q_a)$, for example $\alpha = (a, Q_a)$, to simplify the notation. The assumption that $N_{ab}^c \leq 1$ in the original theory before gauging U(1) implies that fusion multiplicity $N_{\alpha\beta}^\gamma$ of the infinite category $\mathcal{C}'$ is also less than or equal to 1.

We are interested in the category $\tilde{\mathcal{D}}$ obtained from condensing a group of transparent Abelian anyons $\mathcal{T}$ in the infinite category $\mathcal{C}'$. In particular, we focus on the group $\mathcal{T} = \{\tau^k\}_{k\in\mathbb{Z}}$ generated by a single transparent Abelian anyon $\tau$. As discussed in the main text, for the purpose of U(1) gauging, we should condense the group of transparent Abelian anyons $\mathcal{T}$ generated by $\tau = (\nu, \sigma_H)$. In this case, the resulting category $\tilde{\mathcal{D}}$ is the category $\mathcal{D}$ that is the final result of gauging the U(1) symmetry in the MTC $\mathcal{C}$. As we will see later, it is also helpful to consider condensing the group generated by $\tau = (1, s\sigma_H)$ (which is the fusion product of $s$ copies of $(\nu, \sigma_H)$). In this case, the category $\tilde{\mathcal{D}}$ is another intermediate premodular category (with finitely many anyons) towards the final gauged theory $\mathcal{D}$. The following discussion of the condensation of transparent Abelian anyons will be applicable to both cases unless specified otherwise.

The anyons in $\mathcal{C}'$ form orbits under the fusion with $\mathcal{T}$. Let's label these orbits by $[\alpha]$. For each orbit $[\alpha]$, we pick a representative $\alpha \in [\alpha]$. The orbit can be then expressed as $[\alpha] = \{\alpha\tau^k\}_{k\in\mathbb{Z}}$. Since $\tau^k$ for different $k$'s carry different U(1) charges, $\alpha\tau^k \in [\alpha]$ with different $k$ are different anyons. This property has an important consequence that, when we condense the transparent Abelian anyons $\mathcal{T} \subset \mathcal{C}'$ to obtain the category $\tilde{\mathcal{D}}$, the orbits $[\alpha]$ are in one-to-one correspondence to the anyons types in $\tilde{\mathcal{D}}$. Therefore, we will directly use the orbit labels $[\alpha]$ to denote the anyons in $\tilde{\mathcal{D}}$. The fusion rule in $\tilde{\mathcal{D}}$ can be directly obtained from that of the parent category $\mathcal{C}'$:

$$N_{[\alpha][\beta]}^{[\gamma]} = \begin{cases} 1, & \text{if there exists } \Delta_{[\alpha][\beta]}^{[\gamma]} \in \mathcal{T} \text{ such that } N_{\alpha\beta}^{\gamma\left(\Delta_{[\alpha][\beta]}^{[\gamma]}\right)^{-1}} = 1 \text{ in } \mathcal{C}', \\ 0, & \text{otherwise}. \end{cases} \quad (35)$$

Remember the fusion multiplicity in $\mathcal{C}'$ is assumed to be equal to or less than 1. Physically, this fusion rule of $\tilde{\mathcal{D}}$ means that $[\alpha]$ and $[\beta]$ can fuse into $[\gamma]$ so long as their representatives $\alpha$ and $\beta$ can fuse into the representative $\gamma$ up to some condensed transparent anyon $\Delta_{[\alpha][\beta]}^{[\gamma]} \in \mathcal{T}$. $\Delta_{[\alpha][\beta]}^{[\gamma]}$ depends on the choice of representatives of each orbit $[\alpha]$, $[\beta]$ and $[\gamma]$. $\Delta_{[\alpha][\beta]}^{[\gamma]}$ is symmetric under the exchange of its two lower indices, i.e. $\Delta_{[\alpha][\beta]}^{[\gamma]} = \Delta_{[\beta][\alpha]}^{[\gamma]}$.

Now, we calculate the $F$- and $R$-symbols of the category $\tilde{\mathcal{D}}$. As a first step, it is convenient to choose a gauge of the $F$- and $R$-symbols of $\mathcal{C}'$ such that they all take value 1 when restricted to the group of transparent anyons $\mathcal{T}$. Following the discussion of App. C, such a gauge always exists for $\mathcal{T}$ which forms the group $\mathbb{Z}$ under fusion. Let's denote the $F$- and $R$-symbols of $\mathcal{C}'$ in the desired gauge as $F'$ and $R'$. Note that when $\tau = (1, s\sigma_H)$, the $F$- and $R$-symbols in Eq. (21) are already in the desired gauge. However, this is generically not the case when $\tau = (\nu, \sigma_H)$. Hence, additional gauge transformation to Eq. (21) is needed to obtain $F'$ and $R'$ in the desired gauge in the case of $\tau = (\nu, \sigma_H)$.

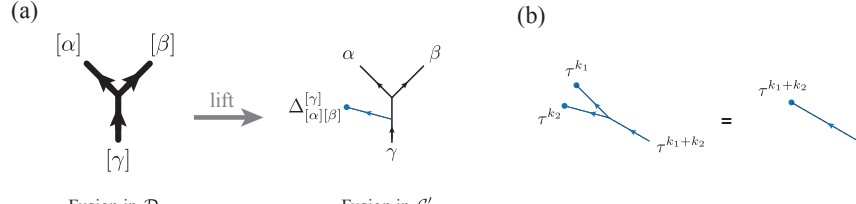

Figure 1: (a) The fusion vertex of the category $\tilde{\mathcal{D}}$ can be lifted into the parent theory $\mathcal{C}'$ as a fusion diagram shown here. (b) For any $k_{1,2} \in \mathbb{Z}$, a single line of condensed transparent anyon $\tau^{k_1+k_2}$ can be split without extra phase given the proper gauge choice of the $F$- and $R$-symbols in $\mathcal{C}'$ as explained in the main text.

The anyon diagrams of the category $\tilde{\mathcal{D}}$ can be lifted back to the anyon diagrams of the parent infinite category $\mathcal{C}'$. In particular, the fusion vertex involving anyons $[\alpha]$, $[\beta]$ and $[\gamma]$ in $\tilde{\mathcal{D}}$ can be expressed as the fusion diagram of $\mathcal{C}'$ shown in Fig. 1 (a). We use thick black line for the anyons in $\tilde{\mathcal{D}}$, thin black lines for anyons in $\mathcal{C}'$ and blue lines for condensed transparent Abelian anyon in $\mathcal{T} \subset \mathcal{C}'$ with the blue dots denoting where the condensation occur. When a generic diagram of $\tilde{\mathcal{D}}$ is lifted to $\mathcal{C}'$, we always adopt the conventions that the all blue lines for the condensed transparent Abelian anyons stay underneath all the black anyon lines and they all condense at the left most part of the diagram. In principle, we should also consider fusing all of the condensed transparent Abelian anyon before their condensation. Due to the gauge choice of $F'$ and $R'$ which are trivial when restricted to the group $\mathcal{T}$ of condensed transparent Abelian anyons, these transparent Abelian anyons can separately condense without extra phase factor associated with the locations of their condensation. Also, this gauge choice allows us to split, without introducing extra phase factor, a condensed anyon line of $\tau^{k_1+k_2} \in \mathcal{T}$ in the way shown in Fig. 1 (b) for any $k_{1,2} \in \mathbb{Z}$.

The $F$-symbol of the category $\tilde{\mathcal{D}}$ can be calculated using the parent category $\mathcal{C}'$. As is shown in Fig. 2, a single $F$-move in the category $\tilde{\mathcal{D}}$ when lifted to the parent category $\mathcal{C}'$ consists of a sequence of $F$- and $R$-moves in $\mathcal{C}'$. Hence, the $F$-symbol $\left(F_{[\delta]}^{[\alpha],[\beta],[\gamma]}\right)_{[\varepsilon],[\varphi]}$ of the category $\tilde{\mathcal{D}}$ can be written in terms of $F'$ and $R'$ of the category $\mathcal{C}'$

$$
\left(F_{[\delta]}^{[\alpha],[\beta],[\gamma]}\right)_{[\varepsilon],[\varphi]} = \left(F'_{\delta'}^{\Delta_{[\alpha][\beta]}^{[\varepsilon]},\varepsilon',\gamma}\right)_{\varepsilon,\delta''} \left(F'_{\delta''}^{\alpha,\beta,\gamma}\right)_{\varepsilon',\varphi'} \left(F'_{\delta}^{\Delta_{[\varepsilon][\gamma]}^{[\delta]},\Delta_{[\alpha][\beta]}^{[\varepsilon]},\delta''}\right)^{-1} \left(F'_{\delta}^{\Delta_{[\alpha][\varphi]}^{[\delta]},\Delta_{[\beta][\gamma]}^{[\varphi]},\delta''}\right)
$$

$$
\times \left(F'_{\delta'''}^{\Delta_{[\beta][\gamma]}^{[\varphi]},\alpha,\varphi'}\right)^{-1} \left(R'^{\Delta_{[\beta][\gamma]}^{[\varphi]},\alpha}\right)^{-1} \left(F'_{\delta'''}^{\alpha,\Delta_{[\beta][\gamma]}^{[\varphi]},\varphi'}\right), \tag{36}
$$

where $\alpha, \beta, \gamma, \varepsilon$ and $\varphi$ are the representatives of their corresponding orbits $[\alpha], [\beta], [\gamma], [\varepsilon]$ and $[\varphi]$. Also, we've defined the following anyon variables

$$
\begin{aligned}
\alpha' &= \Delta_{[\beta][\gamma]}^{[\varphi]} \alpha, & \varepsilon' &= \left(\Delta_{[\alpha][\beta]}^{[\varepsilon]}\right)^{-1} \varepsilon, & \varphi' &= \left(\Delta_{[\beta][\gamma]}^{[\varphi]}\right)^{-1} \varphi, \\
\delta' &= \left(\Delta_{[\varepsilon][\gamma]}^{[\delta]}\right)^{-1} \delta, & \delta'' &= \left(\Delta_{[\alpha][\beta]}^{[\varepsilon]}\right)^{-1} \delta', & \delta''' &= \left(\Delta_{[\alpha][\varphi]}^{[\delta]}\right)^{-1} \delta.
\end{aligned} \tag{37}
$$

On the second row of Fig. 2, we've used the fact that $\Delta_{[\alpha][\beta]}^{[\varepsilon]} \Delta_{[\varepsilon][\gamma]}^{[\delta]} = \Delta_{[\alpha][\varphi]}^{[\delta]} \Delta_{[\beta][\gamma]}^{[\varphi]}$ are the same transparent Abelian anyon. We've suppressed certain indices of the $F$- and $R$-symbol to avoid cluttering in Eq. 36. All the suppressed indices are fully determined via the fusion rule by the explicitly written indices in the same $F$- or $R$-symbols.

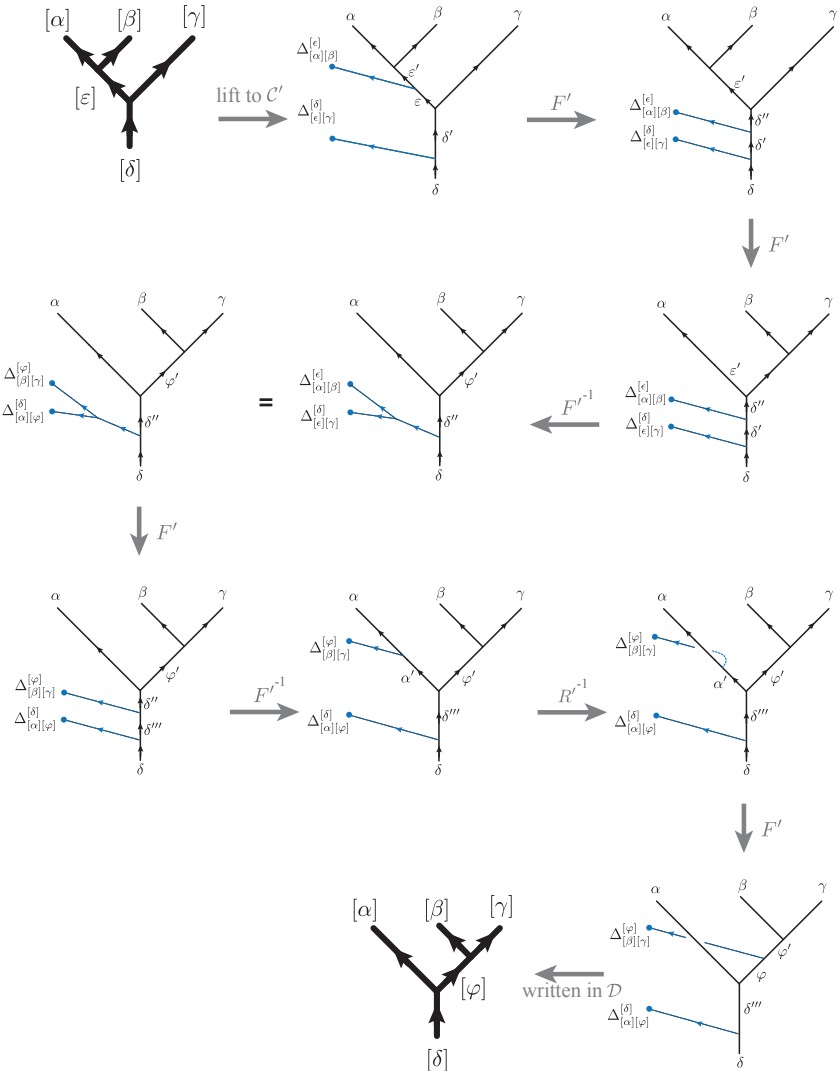

Figure 2: A single $F$-move in the cateory $\tilde{\mathcal{D}}$, when lifted to the parent category $\mathcal{C}'$, consists of a sequence of $F$- and $R$-moves in $\mathcal{C}'$

The $R$-symbol of the category $\tilde{\mathcal{D}}$ can also be obtained by lifting the associated anyon diagram to the parent category $\mathcal{C}'$, as shown in Fig. 3:

$$R_{[\gamma]}^{[\alpha],[\beta]} = R'^{\,\alpha,\beta}_{\gamma'} \, , \tag{38}$$

where $\gamma' = \left( \Delta_{[\alpha][\beta]}^{[\gamma]} \right)^{-1} \gamma$.

We note that similar diagrammatics has been used to compute $F$- and $R$-symbols of symmetry defects in a $G$-crossed braided tensor category [33, 34].

## B.2   Condensation of $\tau = (1, s\sigma_H)$

Now, we consider the case in which the group $\mathcal{T}$ of transparent Abelian anyons is generated by $\tau = (1, s\sigma_H)$. The category obtained from condensing this group $\mathcal{T}$ in $\mathcal{C}'$ will be denoted by $\mathcal{D}_{\text{int}}$. As mentioned earlier, the category $\mathcal{D}_{\text{int}}$ is an intermediate premodular category towards the final theory $\mathcal{D}$ where the U(1) symmetry of the original theory $\mathcal{C}$ is fully gauged.

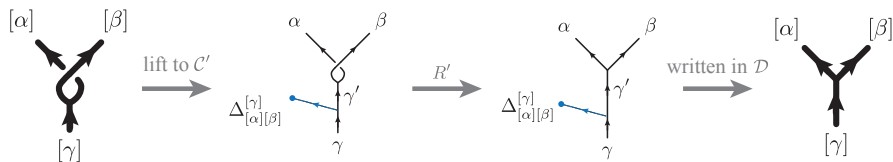

Figure 3: A $R$-move in the category $\tilde{\mathcal{D}}$ can be lifted to a $R$-move in the parent category $\mathcal{C}'$

As commented above, the $F$- and $R$-symbols of $\mathcal{C}'$ given in Eq. (21) is already in the desired gauge, namely their restrictions to the group $\mathcal{T}$ are completely trivial. Under the fusion with transparent anyons in $\mathcal{T}$, each of the orbits in $\mathcal{C}'$ take the form $[(a,Q_a)] = \{(a, Q_a + ks\sigma_H)\}_{k \in \mathbb{Z}}$. Hence, for each orbit $[(a, Q_a)]$, we can choose $(a, \lceil Q_a \rfloor_{|s\sigma_H|}) \in [(a, Q_a)]$ as its representative where $\lceil Q_a \rfloor_{|s\sigma_H|}$ denotes the residue of any $Q_a$ that appears in $[(a, Q_a)]$ modulo $|s\sigma_H|$. $\lceil Q_a \rfloor_{|s\sigma_H|}$ takes value within the interval $[0, |s\sigma_H|)$. When $N^{[(c,Q_c)]}_{[(a,Q_a)][(b,Q_b)]} \neq 0$, U(1) charge conservation together with the fact that $\lceil Q_a \rfloor_{|s\sigma_H|}, \lceil Q_b \rfloor_{|s\sigma_H|}, \lceil Q_c \rfloor_{|s\sigma_H|} \in [0, s\sigma_H)$ leads to the consequence that

$$\Delta^{[(c,Q_c)]}_{[(a,Q_a)][(b,Q_b)]} = (1, \lceil Q_a + Q_b \rfloor_{|s\sigma_H|} - \lceil Q_a \rfloor_{|s\sigma_H|} - \lceil Q_b \rfloor_{|s\sigma_H|}). \tag{39}$$

Now, following App. B.1, we can calculate the $F$- and $R$-symbols of the category $\mathcal{D}_{\text{int}}$ obtained from condensing the group of transparent Abelian anyons $\mathcal{T}$ in the parent category $\mathcal{C}'$. Notice that, the $F$-symbol $\left(F^{(a,Q_a),(b,Q_b),(c,Q_c)}_{(d,Q_d)}\right)_{(e,Q_e),(f,Q_f)}$ of the category $\mathcal{C}'$ shown in Eq. (21) is 1 when any one of $(a, Q_a)$, $(b, Q_b)$, and $(c, Q_c)$ belongs to $\mathcal{T}$. This observation greatly simplifies the expression Eq. (36) and results

$$\left(F^{[(a,Q_a)],[(b,Q_b)],[(c,Q_c)]}_{[(d,Q_d)]}\right)_{[(e,Q_e)],[(f,Q_f)]} = \left(F^{a,b,c}_d\right)_{e,f} e^{\frac{\pi i}{\sigma_H} \lceil Q_a \rfloor_{|s\sigma_H|}(\lceil Q_b + Q_c \rfloor_{|s\sigma_H|} - \lceil Q_b \rfloor_{|s\sigma_H|} - \lceil Q_c \rfloor_{|s\sigma_H|})}. \tag{40}$$

The $R$-symbol of $\mathcal{D}_{\text{int}}$ can be obtained via Eq. (38):

$$R^{[(a,Q_a)],[(b,Q_b)]}_{[(c,Q_c)]} = R^{ab}_c e^{-\frac{\pi i}{\sigma_H} \lceil Q_a \rfloor_{|s\sigma_H|} \lceil Q_b \rfloor_{|s\sigma_H|}}. \tag{41}$$

Here, as a reminder, the $F$-symbol $\left(F^{a,b,c}_d\right)_{e,f}$ and the $R$-symbol $R^{ab}_c$ are those of the original category $\mathcal{C}$.

It is easy to see that the category $\mathcal{D}_{\text{int}}$ is still a premodular category where the anyon represented by $[(v, \sigma_H)]$ is a transparent Abelian anyon. After we further condense the transparent Abelian anyon $[(v, \sigma_H)]$ (and the ones generated by it) in $\mathcal{D}_{\text{int}}$, the resulting category is the category $\mathcal{D}$ that is the final outcome of gauging the U(1) symmetry of the original category $\mathcal{C}$.

## B.3 Application to SU(2)$_k$

Consider a 2+1d U(1)-symmetric topological phase whose topological order is given by the MTC $\mathcal{C} = \text{SU}(2)_k$. There are $k+1$ different types of anyons, labeled by $j = 0, 1/2, \cdots, k/2$. There is a single nontrivial Abelian anyon with $j = k/2$. The fusion rules are given by

$$j_1 \times j_2 = |j_1 - j_2| + \cdots + \min(j_1 + j_2, k - j_1 - j_2). \tag{42}$$

In particular, $\frac{k}{2} \times j = \frac{k}{2} - j$. The Abelian anyon $k/2$ forms a $\mathbb{Z}_2$ group under fusion. The fusion multiplicities of $SU(2)_k$ are given by

$$N_{j_1 j_2}^{j} = \begin{cases} 1, & j \in \{|j_1 - j_2|, \ |j_1 - j_2| + 1, \ ..., \ \min(j_1 + j_2, k - j_1 - j_2)\}, \\ 0, & \text{otherwise}. \end{cases} \tag{43}$$

The $F$-symbols of $SU(2)_k$ are given by [35]

$$\left(F_j^{j_1, j_2, j_3}\right)_{j_{12}, j_{23}} = (-1)^{j_1 + j_2 + j_3 + j} \sqrt{\{2j_{12} + 1\}_q \{2j_{23} + 1\}_q} \begin{bmatrix} j_1 & j_2 & j_{12} \\ j_3 & j & j_{23} \end{bmatrix}_q, \tag{44}$$

where $q = e^{i\frac{2\pi}{k+2}}$, $\{n\}_q \equiv \frac{q^{n/2} - q^{-n/2}}{q^{1/2} - q^{-1/2}}$ and

$$\begin{bmatrix} j_1 & j_2 & j_{12} \\ j_3 & j & j_{23} \end{bmatrix}_q = \Upsilon(j_1, j_2, j_{12})\Upsilon(j_{12}, j_3, j)\Upsilon(j_2, j_3, j_{23})\Upsilon(j_1, j_{23}, j) \tag{45}$$

$$\times \sum_z \left[ \frac{(-1)^z \{z + 1\}_q!}{\{z - j_1 - j_2 - j_{12}\}_q! \{z - j_{12} - j_3 - j\}_q! \{z - j_2 - j_3 - j_{23}\}_q! \{z - j_1 - j_{23} - j\}_q!} \right.$$

$$\left. \times \frac{1}{\{j_1 + j_2 + j_3 + j - z\}_q! \{j_1 + j_{12} + j_3 + j_{23} - z\}_q! \{j_2 + j_{12} + j + j_{23} - z\}_q!} \right].$$

We've used the following definition in Eq. (45),

$$\{n\}_q! \equiv \prod_{m=1}^{n} \{m\}_q,$$

$$\Upsilon(j_1, j_2, j_{12}) \equiv \sqrt{\frac{\{-j_1 + j_2 + j_3\}_q! \{j_1 - j_2 + j_3\}_q! \{j_1 + j_2 - j_3\}_q!}{\{j_1 + j_2 + j_3 + 1\}_q!}}. \tag{46}$$

The summation $\sum_z$ in Eq. (45) runs over all the integer values of $z$ such that the arguments of all $\{\cdot\}_q!$ functions that appear are non-negative.

The $R$-symbol of $SU(2)_k$ is given by

$$R_j^{j_1, j_2} = (-1)^{j - j_1 - j_2} q^{\frac{1}{2}(j(j+1) - j_1(j_1 + 1) - j_2(j_2 + 1))}. \tag{47}$$

Since the anyon $j = k/2$ is the only Abelian anyon in $SU(2)_k$, it should also be identified as the anyon $v$ associated with the $2\pi$ flux (or the vison), i.e. $v = k/2$. Since $\theta_{k/2} = e^{\pi i k/2}$, the Hall conductance satisfies $\sigma_H = \frac{k}{2}$ mod $2\mathbb{Z}$.

When $k$ is odd, the $k/2$ anyon is a semion (or anti-semion) and the whole MTC factorizes into a "$SO(3)_k$" MTC and the semion (or anti-semion) theory. The U(1) symmetry only acts non-trivially on the semion (or anti-semion) sector. Therefore, the gauging of the U(1) symmetry only alters the semion (or anti-semion) sector without changing the "$SO(3)_k$" sector.

The case with even $k$ is more interesting. We will focus on this case in the following. Following the general prescription, we need to introduce the infinite category $\mathcal{C}'$ whose anyons are labeled by $(j, Q)$ where $j$ labels the anyon in $SU(2)_k$ and $Q$ labels the U(1) charge. Since the anyon that corresponds to the $2\pi$ flux is given by $v = k/2$, the U(1) charge of $(j, Q)$ satisfies the constraint that $e^{2\pi i j} = e^{2\pi i Q}$. The $F$- and $R$-symbols of the category $\mathcal{C}'$ can be obtained using Eq. (21), 44 and 47:

$$\left(F_{(j,Q)}'^{(j_1, Q_1), (j_2, Q_2), (j_3, Q_3)}\right)_{(j_{12}, Q_{12}), (j_{23}, Q_{23})} = \left(F_j^{j_1, j_2, j_3}\right)_{j_{12}, j_{23}},$$

$$R_{(j,Q)}'^{(j_1, Q_1), (j_2, Q_2)} = R_j^{j_1, j_2} e^{-\frac{\pi i}{\sigma_H} Q_1 Q_2}. \tag{48}$$

To obtain the MTC $\mathcal{D}$, i.e. the final result of gauging the U(1) symmetry of $\mathcal{C}$, the group $\mathcal{T}$ of transparent Abelian anyon to be condensed is generated by $\tau = (\frac{k}{2}, \sigma_H)$. Note that, $(\frac{k}{2}, -\sigma_H) \in \mathcal{T}$ since the anyon $k/2$ has a $\mathbb{Z}_2$-fusion rule. The anyons in $\mathcal{C}'$ form orbit under fusion with $\mathcal{T}$. One can show that the orbits are completely labeled by $[(j, Q)]$ for $j = 0, 1, 2, ..., k/2$ and $0 \leq Q < |\sigma_H|$. We will use $(j, Q)$ within the same range of $j$ and $Q$ as the representative of its corresponding orbit $[(j, Q)]$. As discussed above, each orbit $[(j, Q)]$ also labels an anyon in the category $\mathcal{D}$ obtained from condensing the group of transparent anyons $\mathcal{T}$.

In the following, we will obtain the data that defines the category $\mathcal{D}$. It is convenient to introduce the function

$$\Lambda(Q_1, Q_2) = \frac{1}{\sigma_H}(Q_1 + Q_2 - \lceil Q_1 + Q_2 \rceil_{|\sigma_H|}), \tag{49}$$

where $\lceil Q \rceil_{|\sigma_H|}$ is the residue of $Q$ modulo $|\sigma_H|$ and it satisfies $0 \leq \lceil Q \rceil_{|\sigma_H|} < |\sigma_H|$. For any $0 \leq Q_1, Q_2 < |\sigma_H|$, the function $\Lambda(Q_1, Q_2)$ only takes value $0$ or $\pm 1$. The fusion rule of category $\mathcal{D}$ is given by

$$[(j_1, Q_1)] \times [(j_2, Q_2)] = \begin{cases} \sum_j N^{j_3}_{j_1 j_2}(j, \lceil Q_1 + Q_2 \rceil_{|\sigma_H|}), & \text{for } \Lambda(Q_1, Q_2) = 0, \\ \sum_{j_3} N^j_{j_1 j_2}(\frac{k}{2} - j, \lceil Q_1 + Q_2 \rceil_{|\sigma_H|}), & \text{for } |\Lambda(Q_1, Q_2)| = 1. \end{cases} \tag{50}$$

This fusion rule and our choice of orbit representatives lead to the function

$$\Delta^{[(j,Q)]}_{[(j_1, Q_1)][(j_2, Q_2)]} = \left(\frac{k}{2}, \sigma_H\right)^{-\Lambda(Q_1, Q_2)}, \tag{51}$$

when the fusion channel $[(j_1, Q_1)] \times [(j_2, Q_2)] \to [(j, Q)]$ exists.

To obtain the $F$-symbol of $\mathcal{D}$, as explained in App. B.1, it is better to start with the preferred gauge choice for the $F$- and $R$-symbols of $\mathcal{C}'$ such that the $F$- and $R$-symbols restricted to $\mathcal{T} \in \mathcal{C}'$ are trivial. Remember that we are currently focusing on the case with even $k$. It turns out that the $F$- and $R$-symbols of $\mathcal{C}'$ obtained in Eq. (48) are already in the preferred gauge choice when $k$ is even. This is due to the fact that $\left(F^{k/2, k/2, k/2}_{k/2}\right)_{0,0} = 1$ for even $k$ according to Eq. (44). Hence, we can proceed to calculate the $F$-symbol of the category $\mathcal{D}$ using Eq. (36) without any extra gauge transformation to Eq. (48) needed. Additional simplification can be achieved by noticing that the $F$-symbol of SU(2)$_k$ given in Eq. (44) has the properties

$$\begin{aligned} \left(F^{j_1 = \frac{k}{2}, j_2, j_3}_j\right)_{\frac{k}{2} - j_2, \frac{k}{2} - j} &= (-1)^{\frac{k}{2} - j_2 - j_3 - j}, \\ \left(F^{j_1, j_2 = \frac{k}{2}, j_3}_j\right)_{\frac{k}{2} - j_1, \frac{k}{2} - j_3} &= (-1)^{\frac{k}{2} - j_1 - j_3 - j}, \\ \left(F^{j_1, j_2, j_3 = \frac{k}{2}}_j\right)_{\frac{k}{2} - j, \frac{k}{2} - j_2} &= (-1)^{\frac{k}{2} - j_1 - j_2 - j}. \end{aligned} \tag{52}$$

With these simplifications taken into account, the $F$-symbol of the MTC $\mathcal{D}$ is given by

$$\begin{aligned} \left(F^{[(j_1, Q_1)], [(j_2, Q_2)], [(j_3, Q_3)]}_{[(j, Q)]}\right)_{[(j_{12}, Q_{12})], [(j_{23}, Q_{23})]} & \\ = (-1)^{(j_{12} - j_3 - j')\Lambda(Q_1, Q_2)}(-1)^{2j(\Lambda(Q_1, Q_2)\Lambda(Q_{12}, Q_3) + \Lambda(Q_1, Q_{23})\Lambda(Q_2, Q_3))} & \quad (53) \\ \times (-1)^{(j_1 \text{sgn} \sigma_H + Q_1)\Lambda(Q_2, Q_3)}\left(F^{j_1, j_2, j_3}_{j''}\right)_{j'_{12}, j'_{23}}, & \end{aligned}$$



with

$$j' = (1 - |\Lambda(Q_{12}, Q_3)|) j + |\Lambda(Q_{12}, Q_3)| \left( \frac{k}{2} - j \right),$$

$$j'' = (1 - |\Lambda(Q_1, Q_2)|) j' + |\Lambda(Q_1, Q_2)| \left( \frac{k}{2} - j' \right),$$

$$j'_{12} = (1 - |\Lambda(Q_1, Q_2)|) j_{12} + |\Lambda(Q_1, Q_2)| \left( \frac{k}{2} - j_{12} \right),$$

$$j'_{23} = (1 - |\Lambda(Q_2, Q_3)|) j_{23} + |\Lambda(Q_2, Q_3)| \left( \frac{k}{2} - j_{23} \right). \tag{54}$$

The $R$-symbol of the MTC $\mathcal{D}$ is given by

$$R^{[(j_1, Q_1)], [(j_2, Q_2)]}_{[(j, Q)]} = R^{j_1, j_2}_{j'} e^{-\frac{\pi i}{\sigma_H} Q_1 Q_2}, \tag{55}$$

where $j' = (1 - |\Lambda(Q_1, Q_2)|) j + |\Lambda(Q_1, Q_2)| \left( \frac{k}{2} - j \right)$.

As an example, we can consider the case of $\mathcal{C} = \text{SU}(2)_2$ with Hall conductance $\sigma_H = 1$, namely $k = 2$ and $\sigma_H = 1$. After gauging the U(1) symmetry, the resulting MTC $\mathcal{D}$ has three anyons $[(0, 0)]$, $[(\frac{1}{2}, \frac{1}{2})]$ and $[(1, 0)]$. The fusion rule given in Eq. (50) matches that of the Ising MTC. One can further check that the $F$- and $R$-symbols of $\mathcal{D}$ obtained from Eq. (54) and Eq. (55) match those of the Ising MTC (up to a gauge transformation).

If we instead consider the case of $\text{SU}(2)_2$ with the Hall conductance $\sigma_H = -1$, namely $k = 2$ and $\sigma_H = -1$. The resulting MTC obtained from gauging the U(1) symmetry becomes the $\text{Spin}(5)_1$ MTC.

## C  Gauge fixing for the group of transparent Abelian anyons

Here, we focus on the $F$- and $R$-symbols of $\mathcal{C}'$ restricted to the group of transparent Abelian anyons $\mathcal{T} = \{\tau^k\}_{k \in \mathbb{Z}}$. Notice that, in the cases we are interested in, $\mathcal{T}$ has a fusion rule isomorphic to the Abelian group $\mathbb{Z}$. The $F$-symbol restricted to $\mathcal{T}$ depends only on the three superscripts and, hence, can be written as $F^{\tau^k, \tau^l, \tau^m}$. The remaining (and suppressed) indices can be inferred from the three superscripts. With the anyon fusion in $\mathcal{T}$ viewed as the group multiplication in $\mathbb{Z}$, we can identify the $F$-symbol restricted to $\mathcal{T}$ as an element in the group cohomology $\mathcal{H}^3[\mathbb{Z}, \text{U}(1)]$, which turns out to be trivial. i.e. $\mathcal{H}^3[\mathbb{Z}, \text{U}(1)] = \mathbb{Z}_1$. Therefore, it is always possible to choose a gauge such that the $F$-symbol restricted to $\mathcal{T}$ is completely trivial.

The $F$- and $R$-symbols restricted to $\mathcal{T}$ must satisfy the hexagon equations. When the $F$-symbols restricted to $\mathcal{T}$ are completely trivial, the hexagon equations read

$$R^{\tau^{k+l}, \tau^m} = R^{\tau^k, \tau^m} R^{\tau^l, \tau^m}, \quad R^{\tau^m, \tau^{k+l}} = R^{\tau^m, \tau^k} R^{\tau^m, \tau^l}. \tag{56}$$

In particular, they imply that $R^{\tau^k, \tau^l} = (R^{\tau, \tau})^{kl} = 1$ because the transparent Abelian anyon $\tau$ under our consideration has a bosonic self-statistics, namely $R^{\tau, \tau} = \theta_\tau = 1$.

## D  Chiral central charge after gauging the U(1) symmetry

Denote by $c'_-$ the chiral central charge of the gauged system. We focus on the case where the Hall conductance $\sigma_H$ is non-zero, i.e. $\sigma_H \neq 0$. We prove below that $c'_- \equiv c_- - \text{sgn} \, \sigma_H \mod 8$, where $c_-$ is the chiral central charge of the original ungauged system.

In a general MTC $\mathcal{C}$, we have the generalized Gauss-Milgram sum:

$$\frac{1}{D_{\mathcal{C}}} \sum_a d_a^2 \theta_a = e^{\frac{2\pi i c_-}{8}} \,, \tag{57}$$

where $d_a$ is the quantum dimension of the anyon $a \in \mathcal{C}$, $D_{\mathcal{C}} \equiv \sqrt{\sum_{a \in \mathcal{C}} d_a^2}$ is the total quantum dimension of the MTC $\mathcal{C}$ and $c_-$ is the chiral central charge of $\mathcal{C}$.

Note that a similar identity holds for a general premodular category $\mathcal{K}$ whose subcategory of transparent anyons $\mathcal{A}$ consists of only Abelian anyons with bosonic self-statistics [36]:

$$\frac{1}{\sqrt{|\mathcal{A}|} D_{\mathcal{K}}} \sum_{a \in \mathcal{K}} d_a^2 \theta_a = e^{\frac{2\pi i c_-}{8}} \,, \tag{58}$$

where $d_a$ is the quantum dimension of the anyon $a \in \mathcal{K}$, $D_{\mathcal{K}} \equiv \sqrt{\sum_{a \in \mathcal{K}} d_a^2}$ is the total quantum dimension of the premodular category $\mathcal{K}$ and $c_-$ is the chiral central charge of the MTC obtained from condensing $\mathcal{A}$ in the premodular category $\mathcal{K}$. Here, $|\mathcal{A}|$ is the number of Abelian anyons in the transparent subcategory $\mathcal{A}$ and is assumed to be finite.

In this work, we start with an U(1)-symmetric MTC $\mathcal{C}$ with a chiral central charge $c_-$ and a Hall conductance $\sigma_H$. Assuming $\sigma_H \neq 0$, we want to calculate the chiral central charge $c_-'$ of the MTC $\mathcal{D}$ obtained from gauging the U(1) symmetry of $\mathcal{C}$. According to the general procedure of gauging presented in App. B, the MTC $\mathcal{D}$ can be viewed as a result of condensing a group of transparent Abelian bosons in the intermediate premodular category $\mathcal{D}_{\text{int}}$ introduced in App. B.2. The intermediate category $\mathcal{D}_{\text{int}}$ is obtained from the infinite category $\mathcal{C}'$ by condensing $(1, s\sigma_H)$, as discussed in App. B.2. In the following, we calculate chiral central charge $c_-'$ of the MTC $\mathcal{D}$ using the intermediate premodular category $\mathcal{D}_{\text{int}}$.

Recall that the simple objects in this premodular category $\mathcal{D}_{\text{int}}$ are labeled by $[(a, Q_a)]$ where $a \in \mathcal{C}$ and the value of $Q_a$ is restricted to the range $0 \leq Q_a < |s\sigma_H|$. Also, we recall the $F$-and $R$- symbols of this category $\mathcal{D}_{\text{int}}$ are given by

$$\left( F_{[(d,Q_d)]}^{[(a,Q_a)],[(b,Q_b)],[(c,Q_c)]} \right)_{[(e,Q_e)],[(f,Q_f)]} = \left( F_d^{a,b,c} \right)_{e,f} e^{\frac{\pi i}{\sigma_H} \lceil Q_a \rfloor_{|s\sigma_H|} (\lceil Q_b + Q_c \rfloor_{|s\sigma_H|} - \lceil Q_b \rfloor_{|s\sigma_H|} - \lceil Q_c \rfloor_{|s\sigma_H|})} \,,$$

$$R_{[(c,Q_c)]}^{[(a,Q_a)],[(b,Q_b)]} = R_c^{ab} e^{-\frac{\pi i}{\sigma_H} \lceil Q_a \rfloor_{|s\sigma_H|} \lceil Q_b \rfloor_{|s\sigma_H|}} \,, \tag{59}$$

where $\left( F_d^{a,b,c} \right)_{e,f}$ and $R_c^{ab}$ are the $F$- and $R$-symbols of the original MTC $\mathcal{C}$ before gauging the U(1) symmetry. The total quantum dimension $D_{\mathcal{D}_{\text{int}}}$ of this premodular category $\mathcal{D}_{\text{int}}$ is given by $D_{\mathcal{D}_{\text{int}}} = \sqrt{|s\sigma_H|} D_{\mathcal{C}}$. In $\mathcal{D}_{\text{int}}$, The quantum dimension $d_{[(a,Q_a)]}$ and the topological twist factor $\theta_{[(a,Q_a)]}$ are given by

$$d_{[(a,Q_a)]} = d_a \,, \quad \theta_{[(a,Q_a)]} = \theta_a e^{-\frac{\pi i}{\sigma_H} \lceil Q_a \rfloor_{|s\sigma_H|}^2} \,, \tag{60}$$

where $d_a$ and $\theta_a$ are the quantum dimension and the topological twist factor of the anyon $a$ in the original MTC $\mathcal{C}$. To obtain the MTC $\mathcal{D}$ from premodular category $\mathcal{D}_{\text{int}}$, one needs to condense the group of transparent Abelian bosons $\mathcal{A} \subset \mathcal{D}_{\text{int}}$ generated by $[(v, \sigma_H)]$. Notice that $|\mathcal{A}| = s$.

Before performing the generalized Gauss-Milgram sum, it is useful to notice that, for a given $a \in \mathcal{C}$, the allowed anyons $[(a, Q_a)] \in \mathcal{D}_{\text{int}}$ have $Q_a \in \{q_a, q_a + 1, ..., q_a + |s\sigma_H| - 1\}$, where $q_a$ is defined via $M_{av} = e^{2\pi i q_a}$ (using the braiding $M_{av}$ of the original MTC $\mathcal{C}$) and

$0 \le q_a < 1$. Therefore, we can write the following generalized Gauss-Milgram sum

$$
\begin{aligned}
e^{\frac{2\pi i c'_-}{8}} &= \frac{1}{\sqrt{|\mathcal{A}|} D_{\mathcal{D}_{\text{int}}}} \sum_{[(a,Q_a)] \in D_{\mathcal{D}_{\text{int}}}} d^2_{[(a,Q_a)]} \theta_{[(a,Q_a)]} \\
&= \frac{1}{s\sqrt{|\sigma_H|} D_{\mathcal{C}}} \sum_{a \in \mathcal{C}} \sum_{k=0}^{s\sigma_H-1} d_a^2 \theta_a e^{-\frac{\pi i}{\sigma_H}(q_a+k)^2}.
\end{aligned}
\tag{61}
$$

First we perform the summation over $k$:

$$
\begin{aligned}
\sum_{k=0}^{s\sigma_H-1} e^{-\frac{i\pi}{\sigma_H}(q_a+k)^2} &= \frac{1}{s} e^{-\frac{\pi i}{\sigma_H} q_a^2} \sum_{k=0}^{s^2\sigma_H-1} e^{-\frac{\pi i}{s^2\sigma_H}(s^2 k^2 + 2q_a s^2 k)} \\
&= \frac{1}{s}\sqrt{|\sigma_H|} e^{-\frac{\pi i}{4}\operatorname{sgn}\sigma_H} \sum_{k=0}^{s^2-1} e^{\frac{\pi i}{s^2}(s^2\sigma_H k^2 + 2q_a s^2 k)} \\
&= \frac{1}{s}\sqrt{|\sigma_H|} e^{-\frac{\pi i}{4}\operatorname{sgn}\sigma_H} \sum_{k=0}^{s^2-1} e^{\pi i(\sigma_H k^2 + 2q_a k)} \\
&= \sqrt{|\sigma_H|} e^{-\frac{\pi i}{4}\operatorname{sgn}\sigma_H} \sum_{k=0}^{s-1} e^{\pi i(\sigma_H k^2 + 2q_a k)} \\
&= \sqrt{|\sigma_H|} e^{\frac{-\pi i}{4}\operatorname{sgn}\sigma_H} \sum_{k=0}^{s-1} \theta_{v^k} M_{v^k,a}.
\end{aligned}
\tag{62}
$$

For the second equality we apply the quadratic reciprocal law for Gauss sums. Now, the full generalized Gauss-Milgram sum of $\mathcal{D}_{\text{int}}$ can be evaluated:

$$
\begin{aligned}
e^{\frac{2\pi i c'_-}{8}} &= \frac{1}{sD_{\mathcal{C}}} e^{-\frac{\pi i}{4}\operatorname{sgn}\sigma_H} \sum_{a \in \mathcal{C}} \sum_{k=0}^{s-1} d_a^2 \theta_a \theta_{v^k} M_{v^k,a} \\
&= \frac{1}{sD_{\mathcal{C}}} e^{-\frac{\pi i}{4}\operatorname{sgn}\sigma_H} \sum_{k=0}^{s-1} \sum_{a \in \mathcal{C}} d^2_{a \times v^k} \theta_{a \times v^k} \\
&= e^{\frac{\pi i}{4}(c_- - \operatorname{sgn}\sigma_H)}.
\end{aligned}
\tag{63}
$$

Hence, we find $c'_- = c_- - \operatorname{sgn}\sigma_H \mod 8$.

# E  General $U(N)_{k,k+Nk'}$ MTC from gauging the U(1) symmetry in $SU(N)_k$

The $U(N)_{k,k+Nk'}$ MTC with $k, k' \in \mathbb{Z}$ and with $k + k'$ an even integer describes the bosonic topological order associated with the 2+1d Chern-Simons theory

$$
\mathcal{L} = -\frac{k}{4\pi} \operatorname{Tr}\left( b\,db - \frac{2i}{3} b^3 \right) - \frac{k'}{4\pi} (\operatorname{Tr} b) d(\operatorname{Tr} b),
\tag{64}
$$

where $b$ is the $U(N)$ gauge connection which is a $N \times N$-matrix-valued 1-form. Let's denote the traceless part of the gauge connection $b$ as the 1-form gauge field $a$ and the trace as a gauge field $A$, i.e. $\operatorname{Tr} b = A$. The gauge field $a$ can be interpreted as an $SU(N)$ gauge connection and the gauge field $A$ as a U(1) gauge connection. When we treat the gauge field $a$ as a dynamical

field while the gauge field $A$ only as a static background gauge field, the Lagrangian in Eq. (64) describes a $\mathcal{C} = \mathrm{SU}(N)_k$ topological order with a U(1) 0-form global symmetry. The associated Hall conductance is given by $\sigma_H = -\frac{k}{N} - k'$. The $2\pi$ flux of the U(1) symmetry is naturally associated with the Abelian anyon $v \in \mathrm{SU}(N)_k$ whose Wilson line generates the $\mathbb{Z}_N$ 1-form symmetry of the $\mathrm{SU}(N)_k$ topological order. This Abelian anyon $v$ has a $\mathbb{Z}_N$ fusion rule (in the $\mathrm{SU}(N)_k$ MTC) and a topological spin $\theta_v = e^{\mathrm{i}2\pi \frac{(N-1)k}{2N}}$. The aforementioned requirement that $k + k'$ is even ensures the consistency condition $e^{\mathrm{i}\pi\sigma_H} = \theta_v$. Obviously, when we gauge this U(1) 0-form global symmetry, we restore the dynamics of the gauge field $A = \mathrm{Tr}\,b$. Hence, the so-obtained topological order after gauging is the $\mathrm{U}(N)_{k,k+Nk'}$ topological order that is described by the Lagrangian Eq. (64) with a fully dynamical the U($N$) gauge field $b$.

The $\mathrm{U}(N)_{k,k+Nk'}$ topological order can also be written as $\mathrm{U}(N)_{k,k+Nk'} = \frac{\mathrm{SU}(N)_k \times \mathrm{U}(1)_{N(k+Nk')}}{\mathbb{Z}_N}$ [24]. Hence, it can be constructed from an anyon condensation in $\mathcal{B}_0 = \mathrm{SU}(N)_k \times \mathrm{U}(1)_{N(k+Nk')}$. The anyon to be condensed here is the composite of $v \in \mathrm{SU}(N)_k$ and the Abelian anyon $x^{-(k+Nk')} \in \mathrm{U}(1)_{N(k+Nk')}$ where $x \in \mathrm{U}(1)_{N(k+Nk')}$ here denotes the anyon that generates the entire $\mathbb{Z}_{N(k+Nk')}$ fusion algebra of the $\mathrm{U}(1)_{N(k+Nk')}$ sector. The Abelian anyon $x^{-(k+Nk')} \in \mathrm{U}(1)_{N(k+Nk')}$ also has a $\mathbb{Z}_N$ fusion rule (in the $\mathrm{U}(1)_{N(k+Nk')}$ MTC) and a topological spin $\theta_{x^{-(k+Nk')}} = e^{\frac{\mathrm{i}2\pi(k+Nk')}{2N}}$. The composite Abelian anyon $(v, x^{-(k+Nk')}) \in \mathcal{B}_0 = \mathrm{SU}(N)_k \times \mathrm{U}(1)_{N(k+Nk')}$ has a bosonic self-statistics and, hence, is allowed to condense.

Similar to the discussion in Sec. 3.2, we first consider the premodular sub-category $\mathcal{B}_1 \subset \mathcal{B}_0 = \mathrm{SU}(N)_k \times \mathrm{U}(1)_{N(k+Nk')}$ that braids trivially with $(v, x^{-(k+Nk')})$. One can show that this sub-category is completely identical to the intermediate premodular category $\mathcal{D}_{\mathrm{int}}$ obtained from applying the general procedures for gauging the U(1) symmetry described in App. B.2 to the $\mathrm{SU}(N)_k$ MTC with a Hall conductance of $\sigma_H = -\frac{k}{N} - k'$. Here, $s = N$ because the vison $v \in \mathrm{SU}(N)_k$ associated with the $2\pi$ flux has a $\mathbb{Z}_N$ fusion rule. Further condensation of $(v, \sigma_H) \in \mathcal{D}_{\mathrm{int}}$ (which is needed for completing the full U(1) gauging procedure described in App. B) in the intermediate category $\mathcal{D}_{\mathrm{int}}$ is equivalent to the condensation of the Abelian anyon $(v, x^{-(k+Nk')})$ in $\mathcal{B}_1$. This condensation yields the $\mathrm{U}(N)_{k,k+Nk'}$ MTC as the final result. The discussion of the $\mathbb{Z}_k$ parafermion in Sec. 3.2 is a special case of the general discussion here.

Also, we notice that when the Hall conductance vanishes, i.e. when $\sigma_H = -\frac{k}{N} - k' = 0$, the field theory Eq. (64) indicates that, after the U(1) symmetry is gauged, the resulting theory is given by $\mathrm{U}(N)_{k,0} = \frac{\mathrm{SU}(N)_k}{\mathbb{Z}_N}$. The theory $\frac{\mathrm{SU}(N)_k}{\mathbb{Z}_N}$ is exactly given by condensing the vison $v$, which generates the $\mathbb{Z}_N$ 1-form symmetry of $\mathrm{SU}(N)_k$, in the ungauged theory $\mathrm{SU}(N)_k$. Remember that $k$ and $k'$ are both integers and $k + k'$ is required to be even. A vanishing Hall conductance only occurs when (1) $k$ is an even-integer multiple of $N$ for even $N$ or (2) $k$ is an integer multiple of $N$ for odd $N$. In these scenarios, the vision $v$ has a bosonic self-statistics $\theta_v = 1$ and is allowed to condense.

# F   Holographic viewpoint of gauging the U(1) symmetry

For a (2+1)d topological order $\mathcal{C}$ with a global (0-form) symmetry $G$, one way to think about the coupling between the (2+1)d topological order to either the background $G$ gauge field or a dynamical $G$-gauge field (after gauging the global symmetry $G$) is via a "holographic" viewpoint in which the (2+1)d spacetime where the (2+1)d topological order resides is the boundary of a (3+1)d spacetime manifold where the $G$ gauge field extends. This viewpoint provides a useful tool in characterizing the possible 't Hooft anomalies of (2+1)d topological order under the a global symmetry $G$ (see Ref. [37–39] for early examples) in which context

the $G$ gauge field is often treated as a static background gauge field. In this appendix, we will discuss such a similar holographic viewpoint for gauging a 0-form global $G$ symmetry of a (2+1)d topological order. In particular, we are interested in the case where $G = U(1)$.

As a start, let's first discus the case when the 0-form global symmetry group $G$ is a finite group. For simplicity, let's consider the $2+1$d topological order $\mathcal{C}$ living on the 2-dimensional spatial $x$-$y$ plane at $z = 0$ while the 3-dimensional bulk is the entire "half-space" with $z \leq 0$. The 1-form $G$ gauge field, background or dynamical, lives in the the entire bulk at $z \leq 0$ and couples to the 2+1d topological order $\mathcal{C}$ on the boundary at $z = 0$. Assuming the global symmetry $G$ of the (2+1)d topological order $\mathcal{C}$ is free of anomaly, we can take (3+1)d bulk to be in a trivial $G$-SPT phase before gauging the global symmetry $G$. After we gauge the symmetry $G$, the bulk hosts a un-twisted dynamical (and deconfined) 1-form $G$ gauge theory, which itself is a (3+1)d topological order with a finite energy gap. What is the (2+1)d topological order $\mathcal{D}$ that is the outcome of gauging the $G$ symmetry of $\mathcal{C}$ purely within (2+1)d? To answer this question in the holographic viewpoint, we can consider the bulk to resides on a "slab of finite width", say the 3d space with $-W \leq z \leq 0$. Here, the $x$-$y$ plane at the boundary at $z = 0$ is still where the original topological order $\mathcal{C}$ resides. On the boundary at $z = -W$, we impose the gapped boundary condition for the (3+1)d dynamical 1-form $G$ gauge theory such that any $G$-flux is allowed to terminate (without energy cost) at the boundary at $z = -W$. Since the entire bulk has a finite width $W$, the total system, including the bulk and the boundaries at $z = 0$ and at $z = -W$, can be viewed as an effective (2+1)d system. This slab construction maintain a finite energy gap at all times and, therefore, yields a (2+1)d topological order that should be identified as $\mathcal{D}$.

Now, let's consider the case of a (2+1)d topological order $\mathcal{C}$ with a global 0-form symmetry $G = U(1)$. We can still consider the (3+1)d bulk defined on the 3D space with $z \leq 0$ and the (2+1)d topological order $\mathcal{C}$ living on the boundary at $z = 0$. Before we gauge the U(1) symmetry, we still require the background ground U(1) 1-form gauge field to be defined in the entire bulk at $z \leq 0$. This background U(1) gauge field couples to the (2+1)d topological order $\mathcal{C}$ at the boundary at $z = 0$. Since the U(1) symmetry of the topological order $\mathcal{C}$ is free of any 't Hooft anomaly, the (3+1)d bulk is in the trivial (3+1)d U(1)-SPT phase. In fact, there is no non-trivial SPT phase with a single 0-form global U(1) symmetry in (3+1)d. Now, let's gauge the U(1) symmetry and, thereby, promoting the background U(1) 1-form gauge field to a dynamical one. We can start by first letting the dynamics of the bulk U(1) gauge field to be governed by the Maxwell theory. Such a bulk theory is obviously gapless. In the bulk, any integer U(1) charge is allowed, while on the boundary at $z = 0$ the infinite set of allowed combinations of anyons in $\mathcal{C}$ and U(1) charges are captured by the infinite category $\mathcal{C}'$ defined via Eq. (18) and 21 assuming $\sigma_H \neq 0$. Directly applying the slab construction to such a bulk theory does not lead any regular (2+1)d topological order (that should have an energy gap and finitely many anyons). We can nevertheless continue to analyze this system.

There is a special excitation on the boundary given by the anyon $(v, \sigma_H) \in \mathcal{C}'$ where $v \in \mathcal{C}$, the vison, is associated with the $2\pi$ flux of the U(1) gauge field. The U(1) charge $\sigma_H$ carried by this anyon $(v, \sigma_H) \in \mathcal{C}'$ is equal to the Hall conductance of the topological order $\mathcal{C}$ on the boundary. When a monopole of the U(1) gauge field, which carries a total of $2\pi$ flux, tunnels through boundary at $z = 0$ into the bulk, it leaves behind on the boundary the composite of the anyon $v \in \mathcal{C}$ and the U(1) charge $\sigma_H$ due to the Hall effect generated by the topological order $\mathcal{C}$ on the boundary. Therefore, the anyon $(v, \sigma_H) \in \mathcal{C}'$ is created when a U(1) monopole tunnels into the bulk. This analysis also suggests that the bulk Maxwell theory has a non-trivial $\theta$-term $\int \frac{\sigma_H}{4\pi} dAdA$ (where $A$ is the 1-form U(1) gauge field).

Now, we condense the U(1) monopoles in the bulk to drive the bulk into a confined phase of the U(1) gauge field. The confined phase is gapped and free of any (3+1)d topological order. The condensation of the U(1) monopoles in the bulk leads to the condensation of the

anyon $(v, \sigma_H) \in \mathcal{C}'$ on the boundary. Therefore, the condensation of the anyon $(v, \sigma_H) \in \mathcal{C}'$ on the boundary yields the topological order $\mathcal{D}$ that is the outcome of gauging the U(1) symmetry of the original topological order $\mathcal{C}$. Here, it may seem that, in the holographic viewpoint, the confinement of the bulk U(1) gauge field via monopole condensation is an extra step beyond simply gauging the U(1) symmetry. We argue that this is natural and necessary because a simple U(1) Maxwell gauge theory in purely (2+1)d does confine automatically. Technically speaking, one should again consider monopole condensation in the bulk that is a slab of finite width, say the 3d space with $-W \leq z \leq 0$. In this case, we can choose the boundary condition at $z = -W$ such that the monopole condensation simply leads to the confinement of the U(1) gauge field both in the bulk and at the boundary $z = -W$. The non-trivial degrees of freedom left all resides on the boundary at $z = 0$ and are captured by the (2+1)d topological order $\mathcal{D}$ (which is obtained from condensing $(v, \sigma_H)$ in the infinite category $\mathcal{C}'$).

So far, we have assumed that $\sigma_H \neq 0$. In fact, one can consider the exact same setup when $\sigma_H = 0$. In this case, when the bulk is governed by the gapless 3+1d Maxwell theory, there are still an infinite set of excitations given by the allowed combinations of anyons in $\mathcal{C}$ and U(1) charges. When the monopoles of $A$ condense driving the 3+1d bulk into the confined phase, the vison $v$, which now carries a vanishing U(1) charge, also condenses on the 2+1d boundary. The condensation of the vison $v$ drives the original MTC $\mathcal{C}$ into a new MTC $\mathcal{D}$ that is equivalent to the result of gauging the U(1) symmetry of the original MTC $\mathcal{C}$.

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
