# Peer review of "Gauging U(1) symmetry in (2+1)d topological phases"

_SciPost Physics, doi:SciPost Phys. 12, 202 (2022)_

## Round 2 · Referee Report · Anonymous · 2022-3-30

Report

This paper is very interesting and should definitely be published.

I would like to point out, however, the relation to the existing literature, which the authors should correct.

The fact that gauging a global symmetry in the 2d WZW model leads to the GKO coset construction was shown in
W. Nahm, Duke Math. J. 54 (1987) 579;
K. Bardakci, E. Rabinovici and B. Saering, Nucl. Phys. B299 (1988) 15;
K. Gawedzki, A. Kupiainen, Nucl.Phys.B 320 (1989) 625,
rather than in reference [27] of this paper.

Instead, reference [27] showed the corresponding description in terms of the 3d Chern-Simons theory. In fact, the construction in [27] is identical to that in this paper.

Finally, the explicit example of parafermions in section III.B of this paper was discussed in appendix C.4 of
N. Seiberg, E. Witten, “Gapped Boundary Phases of Topological Insulators via Weak Coupling”, PTEP 2016 (2016) 12, 12C101, e-Print: 1602.04251 .

  • validity: -
  • significance: -
  • originality: -
  • clarity: -
  • formatting: -
  • grammar: -

Author:  Meng Cheng  on 2022-06-07  [id 2565]

(in reply to Report 1 on 2022-03-30)
Category:
pointer to related literature

We thank the referee for the recommendation, and for the suggestions on connections to earlier literature. We agree with the referee that the papers he/she listed showed the equivalence of gauging 2d WZW and coset construction and have updated the manuscript accordingly.

---

## Round 2 · Referee Report · Anonymous · 2022-4-27

Report

This work describes a general procedure for gauging U(1) symmetries in (2+1)D symmetry-enriched topological phases (SETs) at the level of the unitary modular tensor category describing the topological order. This is a nontrivial contribution to the field; it was well-understood how to systematically derive the gauged theory from the SET data for discrete symmetry groups G, but not for continuous symmetry groups. It is particularly relevant for fractional quantum Hall states, which are the primary examples of SETs that can be realized in experiments; although electronic FQH states are not considered in this paper since they are fermionic SETs, the bosonic case is an important step towards understanding this gauging procedure for fermionic states as well.

The paper was very enjoyable to read – it is very clear and well-written. The only change I would suggest is to add, if possible, a physical argument for the fact that the chiral central charge changes by sgn(\sigma_H); the arguments in both the Abelian and the general case are highly mathematical. This does not affect my recommendation, however, which is that I highly recommend this paper for publication.

  • validity: top
  • significance: high
  • originality: high
  • clarity: top
  • formatting: perfect
  • grammar: perfect

Author:  Meng Cheng  on 2022-06-07  [id 2568]

(in reply to Report 2 on 2022-04-27)
Category:
answer to question

We thank the referee for the encouraging report. Regarding why the chiral central charge changes by sgn(\sigma_H), it is most easily seen from the "hierarchical" description of the gauging, given near the end of Sec. III: the gauged theory is equivalent to $\mathcal{C}\boxtimes \text{U}(1)_{-s^2\sigma_H}|_{(v,s\sigma_H)}$, where $\mathcal{C}$ is the original MTC, $v$ is the vison and $s$ is the order of $v$. Here the subscript $(v,s\sigma_H)$ means the anyon is condensed. From this description, it is clear that the chiral central charge should change by sgn($\sigma_H$), due to the additional U(1) Chern-Simons theory coming from the Hall response of $\mathcal{C}$. While in this work the hierarchy description was introduced mainly as a concise, mathematical description of U(1) gauging, in a follow-up work arXiv:2205.15347 we gave a new interpretation of this description based on symmetry extension and gauging 1-form symmetry, which can be then generalized to gauging other Lie group symmetries as well.

---

## Editorial Decision

published